# Human-Aware Vision-and-Language Navigation: Bridging Simulation to Reality with Dynamic Human Interactions

**Heng Li**[1*]**, Minghan Li**[1*]**, Zhi-Qi Cheng**[1*†]**, Yifei Dong**[2]**,**
**Yuxuan Zhou**[3]**, Jun-Yan He**[4]**, Qi Dai**[5]**, Teruko Mitamura**[1]**, Alexander G. Hauptmann**[1]

[1]Carnegie Mellon University    [2]Columbia University
[3]University of Mannheim    [4]Alibaba Group    [5]Microsoft Research
Project Page: https://lpercc.github.io/HA3D_simulator/

## Abstract

*Vision-and-Language Navigation* (VLN) aims to develop embodied agents that navigate based on human instructions. However, current VLN frameworks often rely on static environments and optimal expert supervision, limiting their real-world applicability. To address this, we introduce *Human-Aware Vision-and-Language Navigation* (HA-VLN), extending traditional VLN by incorporating dynamic human activities and relaxing key assumptions. We propose the *Human-Aware 3D* (HA3D) simulator, which combines dynamic human activities with the Matterport3D dataset, and the *Human-Aware Room-to-Room* (HA-R2R) dataset, extending R2R with human activity descriptions. To tackle HA-VLN challenges, we present the *Expert-Supervised Cross-Modal* (VLN-CM) and *Non-Expert-Supervised Decision Transformer* (VLN-DT) agents, utilizing cross-modal fusion and diverse training strategies for effective navigation in dynamic human environments. A comprehensive evaluation, including metrics considering human activities, and systematic analysis of HA-VLN's unique challenges, underscores the need for further research to enhance HA-VLN agents' real-world robustness and adaptability. Ultimately, this work provides benchmarks and insights for future research on embodied AI and *Sim2Real transfer*, paving the way for more realistic and applicable VLN systems in human-populated environments.

## 1   Introduction

The dream of autonomous robots carrying out assistive tasks, long portrayed in *"The Simpsons,"* is becoming a reality through embodied AI, which enables agents to learn by interacting with their environment [43]. However, effective *Sim2Real* transfer remains a critical challenge [3, 53]. Vision-and-Language Navigation (VLN) [2, 7, 9, 40] has emerged as a key benchmark for evaluating Sim2Real transfer [23], showing impressive performance in simulation [9, 21, 38]. Nevertheless, many VLN frameworks [2, 12, 21, 44, 46, 52] rely on simplifying assumptions, such as *static environments* [25, 39, 50], *panoramic action spaces*, and *optimal expert supervision*, limiting their real-world applicability and often leading to an overestimation of Sim2Real capabilities [51].

To bridge this gap, we propose *Human-Aware Vision-and-Language Navigation* (HA-VLN), extending traditional VLN by incorporating *dynamic human activities* and *relaxing key assumptions*. HA-VLN advances previous frameworks by (1) adopting a limited 60° field-of-view egocentric action space, (2) integrating dynamic environments with 3D human motion models encoded using the SMPL model [31], and (3) learning to navigate considering dynamic environments from suboptimal expert

---

*Equal contribution, authors listed in random order. †Corresponding author. See Author Contributions section for detailed roles (Sec. 5).

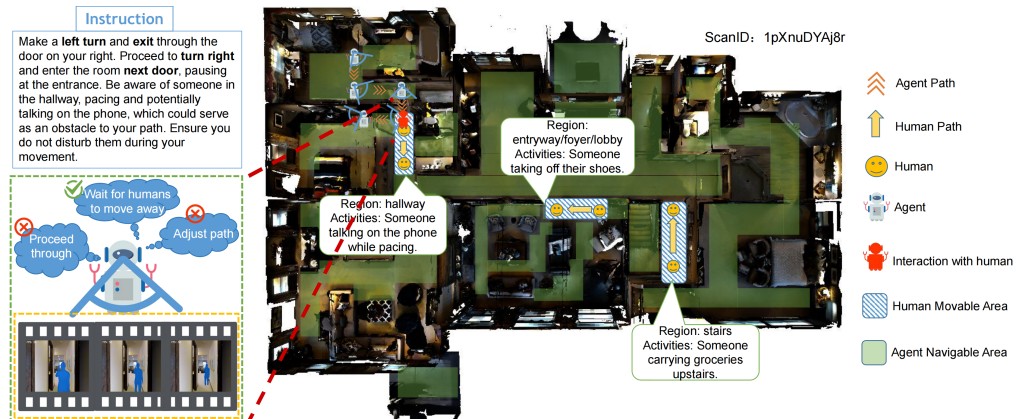

Figure 1: **HA-VLN Scenario:** The agent navigates through environments populated with dynamic human activities. The task involves optimizing routes while maintaining safe distances from humans to address the *Sim2Real* gap. In this scenario, the agent encounters various human activities, such as someone talking on the phone while pacing in the hallway, someone taking off their shoes in the entryway/foyer, and someone carrying groceries upstairs. The HA-VLN agent must adapt its path by waiting for humans to move, adjusting its path, or proceeding through when clear, thereby enhancing real-world applicability.

demonstrations through an adaptive policy (Fig. 6). This setup creates a more realistic and challenging scenario, enabling agents to navigate in human-populated environments while maintaining safe distances, narrowing the gap between simulation and real-world scenes.

To support HA-VLN research, we introduce the Human-Aware 3D (HA3D) simulator, a realistic environment combining dynamic human activities with the Matterport3D dataset [6]. HA3D utilizes the self-collected Human Activity and Pose Simulation (HAPS) dataset, which includes 145 human activity descriptions converted into 435 detailed 3D human motion models using the SMPL model [31] (Sec. 2.1). The simulator provides an interactive annotation tool for placing human models in 29 different indoor areas across 90 building scenes (Fig. 12). Moreover, we introduce the Human-Aware Room-to-Room (HA-R2R) dataset, an extension of the Room-to-Room (R2R) dataset [2] incorporating human activity descriptions. HA-R2R includes 21,567 instructions with an expanded vocabulary and activity coverage compared to R2R (Fig. 3 and Sec. 2.2).

Building upon the HA-VLN task and the HA3D simulator, we propose two multimodal agents to address the challenges posed by dynamic human environments: the Expert-Supervised Cross-Modal (VLN-CM) agent and the Non-Expert-Supervised Decision Transformer (VLN-DT) agent. The innovation of these agents lies in their cross-modal fusion module, which dynamically weights language and visual information, enhancing their understanding and utilization of different modalities. VLN-CM learns by imitating expert demonstrations (Sec. 2.2), while VLN-DT demonstrates the potential to learn solely from random trajectories without expert supervision (Fig. 4, right). We also design a rich reward function to incentivize agents to navigate effectively (Fig. 5).

To comprehensively evaluate the performance of the HA-VLN task, we design new metrics considering human activities, and highlight the unique challenges faced by HA-VLN (Sec. 3.2). Evaluating state-of-the-art VLN agents on the HA-VLN task reveals a significant performance gap compared to the Oracle, even after retraining, thereby underscoring the complexity of navigating in dynamic human environments (Sec. 3.3). Moreover, experiments show that VLN-DT, trained solely on random data, achieves performance comparable to VLN-CM under expert supervision, thus demonstrating its superior generalization ability (Sec. 3.4). Finally, we validate the agents in the real world using a quadruped robot, exhibiting perception and avoidance capabilities, while also emphasizing the necessity of further improving real-world robustness and adaptability (Sec. 3.5).

Our main contributions are as follows: (1) Introducing HA-VLN, a new task that extends VLN by incorporating dynamic human activities and relaxing assumptions; (2) Offering HA3D, a realistic simulator, and HA-R2R, an extension of the R2R dataset, to support HA-VLN research and enable the development of robust navigation agents; (3) Proposing VLN-CM and VLN-DT agents that utilize expert and non-expert supervised learning to address the challenges of HA-VLN, showcasing the effectiveness of cross-modal fusion and diverse training strategies; and (4) Designing comprehensive evaluations for HA-VLN, providing benchmarks and insights for future research.

## 2 Human-Aware Vision-and-Language Navigation

We introduce *Human-Aware Vision-and-Language Navigation* (*HA-VLN*), an extension of traditional Vision-and-Language Navigation (*VLN*) that bridges the *Sim2Real gap* [3, 23, 53] between simulated and real-world navigation scenarios. As shown in Fig. 1, *HA-VLN* involves an embodied agent navigating from an initial position to a target location within a dynamic environment, guided by natural language instructions $\mathcal{I} = \langle w_1, w_2, \ldots, w_L \rangle$, where $L$ denotes the total number of words and $w_i$ represents an individual word. At the beginning of each episode, the agent assesses its initial state $\mathbf{s}_0 = \langle \mathbf{p}_0, \phi_0, \lambda_0, \Theta_0^{60} \rangle$ within a $\Delta t = 2$ seconds observation window, where $\mathbf{p}_0 = (x_0, y_0, z_0)$ represents the initial 3D position, $\phi_0$ the heading, $\lambda_0$ the elevation, and $\Theta_0^{60}$ the egocentric view within a 60-degree field of view. The agent executes a sequence of actions $\mathcal{A}_T = \langle a_0, a_1, \ldots, a_T \rangle$, resulting in states and observations $\mathcal{S}_T = \langle \mathbf{s}_0, \mathbf{s}_1, \ldots, \mathbf{s}_T \rangle$, where each action $a_t \in \mathcal{A} = \langle a_{\text{forward}}, a_{\text{left}}, a_{\text{right}}, a_{\text{up}}, a_{\text{down}}, a_{\text{stop}} \rangle$ leads to a new state $\mathbf{s}_{t+1} = \langle \mathbf{p}_{t+1}, \phi_{t+1}, \lambda_{t+1}, \Theta_{t+1}^{60} \rangle$. The episode concludes with the stop action $a_{\text{stop}}$.

In contrast to traditional *VLN* tasks [2, 13, 26, 40, 45], *HA-VLN* addresses the *Sim2Real gap* [3, 23, 53] by relaxing three key assumptions, as depicted in Fig. 1:

1. **Egocentric Action Space:** *HA-VLN* employs an egocentric action space $\mathcal{A}$ with a limited $60°$ field of view $\Theta_t^{60}$, requiring the agent to make decisions based on human-like visual perception. The state $\mathbf{s}_t = \langle \mathbf{p}_t, \phi_t, \lambda_t, \Theta_t^{60} \rangle$ captures the agent's egocentric perspective at time $t$, enabling effective navigation in real-world scenarios.

2. **Dynamic Environments:** *HA-VLN* introduces dynamic environments based on 3D human motion models $\mathbf{H} = \langle h_1, h_2, \ldots, h_N \rangle$, where each frame $h_i \in \mathbb{R}^{6890 \times 3}$ encodes human positions and shapes using the Skinned Multi-Person Linear (SMPL) model [31]. The agent must perceive and respond to these activities in real-time while maintaining a safe distance $d_{\text{safe}}$, reflecting real-world navigation challenges.

3. **Sub-optimal Expert Supervision:** In *HA-VLN*, agents learn from sub-optimal expert demonstrations that provide navigation guidance accounting for the dynamic environment. The agent's policy $\pi_{\text{adaptive}}(a_t | \mathbf{s}_t, \mathcal{I}, \mathbf{H})$ aims to maximize the expected reward $\mathbb{E}[r(\mathbf{s}_{t+1})]$, considering human interactions and safe navigation. The reward function $r : \mathcal{S} \to \mathbb{R}$ assesses the quality of navigation at each state, allowing better handling of imperfect instructions in real-world tasks.

Building upon these relaxed assumptions, a key feature of *HA-VLN* is the inclusion of human activities captured at 16 FPS. When human activities fall within the agent's field of view $\Theta_t^{60}$, the agent is considered to be interacting with humans. *HA-VLN* introduces the Adaptive Response Strategy, where the agent detects and responds to human movements, anticipating trajectories and making real-time path adjustments. Formally, this strategy is defined as:

$$\pi_{\text{adaptive}}(a_t | \mathbf{s}_t, \mathcal{I}, \mathbf{H}) = \arg \max_{a_t \in \mathcal{A}} P(a_t | \mathbf{s}_t, \mathcal{I}) \cdot \mathbb{E}[r(\mathbf{s}_{t+1})], \tag{1}$$

where $\mathbb{E}[r(\mathbf{s}_{t+1})]$ represents the expected reward considering human interactions and safe navigation. To support the agent in learning, the *HA3D* simulator (Sec. 2.1) provides interfaces to access human posture, position, and trajectories, while *HA-VLN* employs sub-optimal expert supervision (Sec. 2.2) to provide weak signals, reflecting real-world scenarios with imperfect demonstration.

### 2.1 HA3D Simulator: Integrating Dynamic Human Activities

The *Human-Aware 3D (HA3D) Simulator* generates dynamic environments by integrating natural human activities from the custom-collected *Human Activity and Pose Simulation (HAPS) dataset* with the photorealistic environments of the Matterport3D dataset [6] (see Fig. 2 and Fig. 12).

**HAPS Dataset.** HAPS addresses the limitations of existing human motion datasets by identifying 29 distinct indoor regions across 90 architectural scenes and generating 145 human activity descriptions. These descriptions, validated through human surveys and quality control using GPT-4 [5], encompass realistic actions such as walking, sitting, and using a laptop. The Motion Diffusion Model (MDM) [17] converts these descriptions into 435 detailed 3D human motion models $\mathbf{H}$ using the SMPL model, with each description transformed into three distinct 120-frame motion sequences[1]. The dataset also

---

[1] $\mathbf{H} \in \mathbb{R}^{435 \times 120 \times (10 + 72 + 6890 \times 3)}$, representing 435 models, 120 frames each, with shape, pose, and mesh vertex parameters. See *Realistic Human Rendering* for more details.

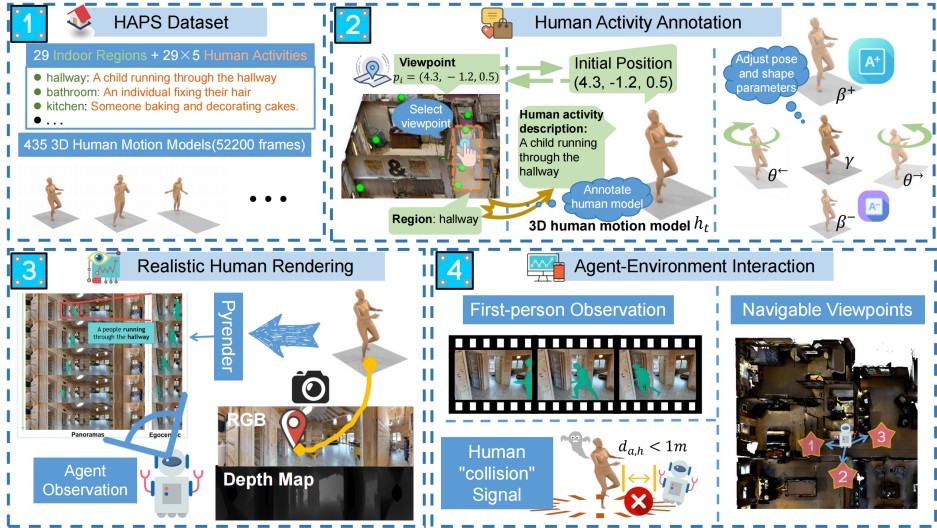

**Figure 2: Human-Aware 3D (HA3D) Simulator Annotation Process:** HA3D integrates dynamic human activities from the Human Activity and Pose Simulation (HAPS) dataset into the photorealistic environments of Matterport3D. The annotation process involves: (1) integrating the HAPS dataset, which includes 145 human activity descriptions converted into 435 detailed 3D human motion models in 52,200 frames; (2) annotating human activities within various indoor regions across 90 building scenes using an interactive annotation tool; (3) rendering realistic human models; and (4) enabling interactive agent-environment interactions.

includes annotations of human-object interactions and the relationship between human activities and architectural layouts. After manual selection, approximately 422 models were retained. Further details on the dataset are provided in App. B.1.

**Human Activity Annotation.** An interactive annotation tool accurately locates humans in different building regions (see Fig. 12). Users explore buildings, select viewpoints $\mathbf{p}_i = (x_i, y_i, z_i)$, set initial human positions, and choose 3D human motion models $\mathbf{H}_i$ based on the environment of $\mathbf{p}_i$. To follow real-world scenarios, multiple initial human viewpoints $\mathbf{P}_{\text{random}} = \{\mathbf{p}_1, \mathbf{p}_2, \ldots, \mathbf{p}_k\}$ are randomly selected from a subset of all viewpoints in the building. The number of people in each building is estimated by dividing the building area by the average area per capita in the U.S. (2021, $67m^2$) [35] and rounding up. In the Matterport3D dataset, these viewpoints are manually annotated to facilitate the transfer from other VLN tasks to HA-VLN. This setup ensures agents can navigate environments with dynamic human activities updated at 16 FPS, allowing real-time perception and response. Detailed statistics of activity annotation are in App. B.2.

**Realistic Human Rendering.** HA3D employs *Pyrender* to render dynamic human bodies with high visual realism. The rendering process aligns camera settings with the agent's perspective and integrates dynamic human motion using a 120-frame SMPL mesh sequence $\mathbf{H} = \langle h_1, h_2, \ldots, h_{120} \rangle$. Each frame $h_t = (\beta_t, \theta_t, \gamma_t)$ consists of shape parameters $\beta_t \in \mathbb{R}^{10}$, pose parameters $\theta_t \in \mathbb{R}^{72}$, and mesh vertices $\gamma_t \in \mathbb{R}^{6890 \times 3}$ calculated based on $\beta_t$ and $\theta_t$ through the SMPL model. At each time step, the 3D mesh $h_t$ is dynamically generated, with vertices $\gamma_t$ algorithmically determined to form the human model accurately. These vertices are then used to generate depth maps $\mathbf{D}_t$, distinguishing human models from other scene elements. HA3D allows real-time adjustments of human body parameters, enabling the representation of diverse appearances and enhancing interactivity. More details on the rendering pipeline and examples of rendered human models are in App. B.3.

**Agent-Environment Interaction.** Compatible with the Matterport3D simulator's configurations [2], HA3D provides agents with environmental feedback signals at each time step $t$, including first-person RGB-D video observation $\Theta_t^{60}$, navigable viewpoints, and a human "collision" feedback signal $c_t$. The agent receives its state $\mathbf{s}_t = \langle \mathbf{p}_t, \phi_t, \lambda_t, \Theta_t^{60} \rangle$, where $\mathbf{p}_t = (x_t, y_t, z_t)$, $\phi_t$, and $\lambda_t$ denote position, heading, and elevation, respectively. The agent's policy $\pi_{\text{adaptive}}(a_t | \mathbf{s}_t, \mathcal{I}, \mathbf{H})$ maximizes expected reward $\mathbb{E}[r(\mathbf{s}_{t+1})]$ by considering human interactions for safe navigation. The collision feedback signal $c_t$ is triggered when the agent-human distance $d_{a,h}(t)$ falls below a threshold $d_{\text{threshold}}$. Customizable collision detection and feedback parameters enhance agent-environment interaction. Details on visual feedback, optimization, and extended interaction capabilities are in App. B.4.

**Implementation and Performance.** Developed using C++/Python, OpenGL, and Pyrender, HA3D integrates with deep learning frameworks like PyTorch and TensorFlow. It offers flexible configuration options, achieving up to 300 fps on an NVIDIA RTX 3050 GPU with 640x480 resolution. Running on Linux, the simulator has a low memory usage of 40MB and supports multi-processing for parallel execution of simulation tasks. Its modular architecture enables easy extension and customization. The simulator supports various rendering techniques, enhancing visual realism. It provides high-level APIs for real-time data streaming and interaction with external controllers. PyQt5-based annotation tools with an intuitive interface will be made available to researchers. Additional details on the simulator's implementation, performance, and extensibility are provided in App. B.5.

## 2.2 Human-Aware Navigation Agents

We introduce the *Human-Aware Room-to-Room (HA-R2R) dataset*, extending the Room-to-Room (R2R) dataset [2] by incorporating human activity descriptions to create a more realistic and dynamic navigation environment. To address HA-VLN challenges, we propose two agents: the *expert-supervised Cross Modal (VLN-CM) agent* and *the non-expert-supervised Decision Transformer (VLN-DT) agent*. An *Oracle agent* provides ground truth supervision for training and benchmarking.

**HA-R2R Dataset.** HA-R2R extends R2R dataset by incorporating human activity annotations while preserving its fine-grained navigation properties.The dataset was constructed in two steps: 1) mapping R2R paths to the HA3D simulator, manually annotating human activities at key locations; and 2) using GPT-4 [1] to generate new instructions by combining original instructions, human activity descriptions, and relative position information, followed by human validation. The resulting dataset contains 21,567 human-like instructions with 145 activity types, categorized as *start* (1,047), *obstacle* (3,666), *surrounding* (14,469), and *end* (1,041) based on their positions relative to the agent's starting point (see App. C.1 for details). Compared to R2R, HA-R2R's average instruction length increased from 29 to 69 words, with the vocabulary expanding from 990 to 4,500. Fig. 3A shows the instruction length distribution by activity count, while Fig. 3B compares HA-R2R and R2R distributions. Fig. 3C summarizes viewpoints affected by human activities, and Fig. 14 illustrates the instruction quality by analyzing common word frequencies. More details are provided in App. C.1.

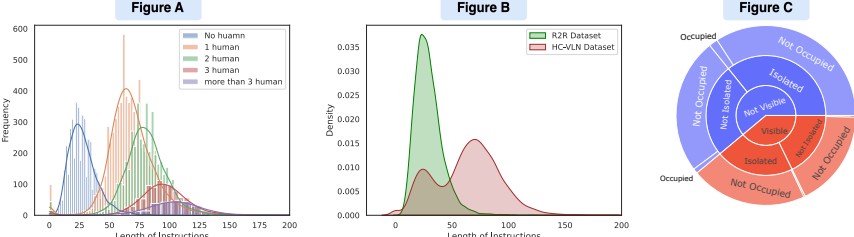

Figure 3: **Dataset Analysis of HA-R2R: (A)** Impact of human activities on instruction length, tokenized using NLTK WordNet, showing the variation in instruction length caused by different types of human activities. **(B)** Comparison of instruction length distributions between HA-R2R and the original R2R dataset. HA-R2R demonstrates a more uniform distribution, facilitating balanced training. **(C)** Analysis of viewpoints affected by human activities: "Visible" denotes activities within the agent's sight, "Isolated" refers to key navigation nodes impacted by human activities, and "Occupied" indicates the presence of humans at specific viewpoints.

**Oracle Agent: Ground Truth Supervision.** The Oracle agent serves as the ground truth supervision source to guide and benchmark the training of expert-supervised and non-expert-supervised agents in the HA-VLN system. Designed as a *teacher*, the Oracle provides realistic supervision derived from the HA-R2R dataset, strictly following language instructions while dynamically avoiding human activities along navigation paths to ensure maximal expected rewards. Let $G = (N, E)$ be the global navigation graph, with nodes $N$ (locations) and edges $E$ (paths). When human activities affect nodes $n \in N$ within radius $r$, those nodes form subset $N_h$. The Oracle's policy $\pi^*_{\text{adaptive}}$ re-routes on the modified graph $G' = (N \setminus N_h, E')$, where $E'$ only includes edges avoiding $N_h$, ensuring the Oracle avoids human-induced disturbances while following navigation instructions optimally. Algorithm 1 details the Oracle's path planning and collision avoidance strategies. During training, at step $t$, a cross-entropy loss maximizes the likelihood of true target action $a^*_t$ given the previous state-action sequence $\langle s_0, a_0, s_1, a_1, \ldots, s_t \rangle$. The target output $a^*_t$ is defined as the Oracle's next action from the current location to the goal. Please refer to App. C.2 for more details.

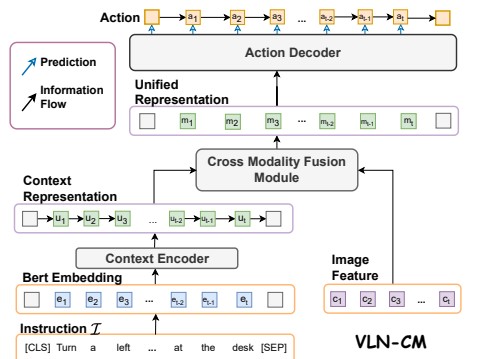 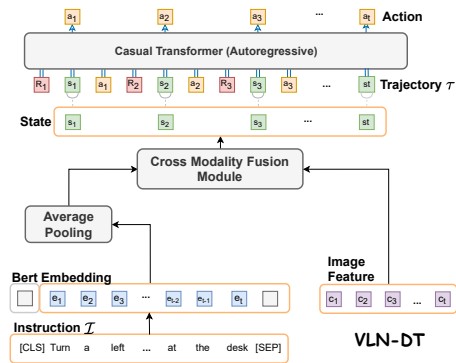

Figure 4: **Model Architectures of Navigation Agents:** The architectures of the Vision-Language Navigation Cross-Modal (VLN-CM) agent (left) and the Vision-Language Navigation Decision Transformer (VLN-DT) agent (right). Both agents employ a cross-modality fusion module to effectively integrate visual and linguistic information for predicting navigation actions. VLN-CM utilizes an LSTM-based sequence-to-sequence model for expert-supervised learning, while VLN-DT leverages an autoregressive transformer model to learn from random trajectories without expert supervision.

**VLN-CM: Multimodal Integration for Supervised Learning.** We propose the Vision-Language Navigation Cross-Modal (VLN-CM) agent, an LSTM-based sequence-to-sequence model [2] augmented with a cross modality fusion module for effective multimodal integration (Fig. 4, left). The language instruction $\mathcal{I} = \langle w_1, w_2, \cdots, w_L \rangle$, where $w_i$ denotes the $i$-th word, is encoded into BERT embeddings [11] $\{e_1, e_2, \cdots, e_L\}$, which are processed by an LSTM to yield context-aware representations $\{u_1, u_2, \cdots, u_L\}$. Simultaneously, visual observations $\Theta_t$ at each timestep $t$ are encoded using ResNet-152 [19], producing an image feature map $\{c_1, c_2, \cdots, c_N\}$, where $N$ is the number of visual features. The fusion module integrates the context encoder outputs and image features via cross-attention, generating a unified representation $m_t$ at each timestep $t$. An LSTM-based action decoder predicts the next action $a_{t+1}$ from the action space $\mathcal{A} = \{a_{\text{forward}}, a_{\text{left}}, a_{\text{right}}, a_{\text{up}}, a_{\text{down}}, a_{\text{stop}}\}$ conditioned on $m_t$ and the previous action $a_t$. The agent is trained via supervised learning from an expert Oracle agent using cross-entropy loss:

$$\mathcal{L}_{\text{CE}} = \sum_{a \in \mathcal{A}} y_t(a) \log p(a_t | \mathcal{I}, \Theta_t), \tag{2}$$

where $\mathcal{L}_{\text{CE}}$ is the cross-entropy loss, $y_t(a)$ is the ground truth action distribution from the expert trajectory at timestep $t$, and $p(a_t | \mathcal{I}, \Theta_t)$ is the predicted action distribution given instruction $\mathcal{I}$ and observation $\Theta_t$ at timestep $t$.

**VLN-DT: Reinforcement Learning with Decision Transformers.** We present the Vision-Language Navigation Decision Transformer (VLN-DT), an autoregressive transformer [8, 41] with a cross-modality fusion module for navigation without expert supervision[2] (Fig. 4, right). VLN-DT learns from sequence representations $\tau = (\hat{G}_1, \mathbf{s}_1, \mathbf{a}_1, \ldots, \hat{G}_t, \mathbf{s}_t)$ to predict the next action $\mathbf{a}_t \in \mathcal{A}$, where $\mathbf{s}_t$ is the state at timestep $t$, and $\hat{G}_t = \sum_{t'=t}^{T} r_{t'}$ is the Return to Go. The cross-modality fusion module computes $\mathbf{s}_t$ by processing the average pooling vector of the BERT embedding [11] for a language instruction $\mathcal{I}$ (excluding the [CLS] token) and the image feature map of the current observation $\Theta_t^{60}$, extracted using a pre-trained ResNet-152 [19]. The fusion module dynamically weights the language and visual modalities using an attention mechanism, enhancing $\mathbf{s}_t$. The fused representations are then fed into the causal transformer, which models $\tau$ autoregressively to determine $\mathbf{a}_t$. We train VLN-DT using $10^4$ random walk trajectories, each with a maximum length of 30 steps, a context window size of 15 steps, and an initial Return To Go of 5 to guide the agent's exploration-exploitation balance [8]. Three reward types are designed to incentivize effective navigation: target reward (*based on distance to the target*), distance reward (*based on movement towards the target*), and human reward (*based on collisions with humans*) [2, 26, 40]. Fig. 5 shows the impact of different reward strategies on navigation performance. The loss function $\mathcal{L}_{\text{CE}}$ for training VLN-DT is a supervised learning objective with cross-entropy loss:

$$\mathcal{L}_{\text{CE}} = \sum_{a \in \mathcal{A}} y_t^*(a) \log p(a_t | s_t), \tag{3}$$

---

[2]"Without Expert Supervision" means training with random trajectories instead of expert ones.

Table 1: Egocentric vs. Panoramic Action Space Comparison

| Action Space | Validation Seen | | | | Validation Unseen | | | |
|---|---|---|---|---|---|---|---|---|
| | NE ↓ | TCR ↓ | CR ↓ | SR ↑ | NE ↓ | TCR ↓ | CR ↓ | SR ↑ |
| **Egocentric** | 7.21 | 0.69 | 1.00 | 0.20 | 8.09 | 0.54 | 0.58 | 0.16 |
| **Panoramic** | 5.58 | 0.24 | 0.80 | 0.34 | 7.16 | 0.25 | 0.57 | 0.23 |
| **Difference** | -1.63 | -0.45 | -0.20 | +0.14 | -0.93 | -0.29 | -0.01 | +0.07 |
| **Percentage** | -22.6% | -65.2% | -20.0% | +70.0% | -11.5% | -53.7% | -1.7% | +43.8% |

Table 2: Optimal vs. Sub-Optimal Expert Comparison

| Expert Type | Validation Seen | | | | Validation Unseen | | | |
|---|---|---|---|---|---|---|---|---|
| | NE ↓ | TCR ↓ | CR ↓ | SR ↑ | NE ↓ | TCR ↓ | CR ↓ | SR ↑ |
| **Optimal** | 3.61 | 0.15 | 0.52 | 0.53 | 5.43 | 0.26 | 0.69 | 0.41 |
| **Sub-optimal** | 3.98 | 0.18 | 0.63 | 0.50 | 5.24 | 0.24 | 0.67 | 0.40 |
| **Difference** | +0.37 | +0.03 | +0.11 | -0.03 | -0.19 | -0.02 | -0.02 | -0.01 |
| **Percentage** | +10.2% | +20.0% | +21.2% | -5.7% | -3.5% | -7.7% | -2.9% | -2.4% |

where $y_t^*(a)$ is the ground truth action distribution from the random trajectory at timestep $t$, and $p(a_t|s_t)$ is the predicted action distribution given instruction $\mathcal{I}$ and observation $\Theta_t$ at timestep $t$. The implementation of VLN-DT is summarized in App. C.3.

# 3  Experiments

We evaluated our Human-Aware Vision-and-Language Navigation (HA-VLN) task, focusing on human perception and navigation. Experiments included assessing different assumptions (Sec. 3.2), comparing with state-of-the-art (SOTA) VLN agents (Sec. 3.3)[3], analyzing our agents' performance (Sec. 3.4), and validating with real-world quadruped robot tests (Sec. 3.5).

## 3.1  Evaluation Protocol for HA-VLN Task

We propose a two-fold evaluation protocol for the HA-VLN task, focusing on both *human perception* and *navigation* aspects. The *human perception* metrics evaluate the agent's ability to perceive and respond to human activities, while the *navigation-related* metrics assess navigation performance. As human activities near critical nodes[4] greatly influence navigation, we introduce a strategy to handle dynamic human activities for more accurate evaluation[5]. Let $A^c i$ be the set of human activities at critical nodes in navigation instance $i$. The updated *human perception* metrics are:

$$TCR = \frac{\sum_{i=1}^{L}(c_i - |A^c i|)}{L}, \quad CR = \frac{\sum_{i=1}^{L} \min{(c_i - |A^c i|, 1)}}{\beta L}, \tag{4}$$

where *TCR* reflects the overall frequency of the agent colliding with human-occupied areas within a 1-meter radius, *CR* is the ratio of navigation instances with at least one collision, and $\beta$ denotes the ratio of instructions affected by human activities. The updated *navigation* metrics are:

$$NE = \frac{\sum_{i=1}^{L} di}{L}, \quad SR = \frac{\sum_{i=1}^{L} \mathbb{I}\left(c_i - |A_i^c| = 0\right)}{L}, \tag{5}$$

where *NE* is the distance between the agent's final position and the target location, and *SR* is the proportion of navigation instructions successfully completed without collisions and within a predefined navigation range. Please refer to App. D.1 for more details.

## 3.2  Evaluating HA-VLN Assumptions

We assessed the impact of relaxing traditional assumptions on navigation performance by comparing HA-VLN and VLN task, relaxing each assumption individually.

**Panoramic vs. Egocentric Action Space** (Tab. 1): Shifting from a panoramic to an egocentric action space significantly degrades overall performance, with Success Rate (SR) dropping by 70.0% in seen environments and by 43.8% in unseen environments. Additionally, there is a marked increase in

---

[3]We report performance of SOTA agents in traditional VLN – Room-to-Room(R2R)[2] for comparison.
[4]Node $n_i$ is critical if $U(n_i) \cap U(n_{i+1}) = \emptyset$, where $U(n_i)$ is the set of nodes reachable from $n_i$.
[5]Ignoring activities at critical nodes decreases *CR* and *TCR*, while increasing *SR*.

Table 4: Performance of SOTA VLN Agents on HA-VLN (Retrained)

| Method | Validation Seen | | | | | | Validation Unseen | | | | | |
|---|---|---|---|---|---|---|---|---|---|---|---|---|
| | w/o human | | w/ human | | Difference | | w/o human | | w/ human | | Difference | |
| | NE ↓ | SR ↑ | NE ↓ | SR ↑ | NE | SR | NE ↓ | SR ↑ | NE ↓ | SR ↑ | NE | SR |
| Speaker-Follower [12] | 6.62 | 0.35 | 5.58 | 0.34 | -15.7% | -2.9% | 3.36 | 0.66 | 7.16 | 0.23 | +113.1% | -65.2% |
| Rec (PREVALENT) [21] | 3.93 | 0.63 | 4.95 | 0.41 | +25.9% | -34.9% | 2.90 | 0.72 | 5.86 | 0.36 | +102.1% | -50.0% |
| Rec (OSCAR) [21] | 4.29 | 0.59 | 4.67 | 0.42 | +8.9% | -28.8% | 3.11 | 0.71 | 5.86 | 0.38 | +88.4% | -46.5% |
| Airbert [16] | 4.01 | 0.62 | 3.98 | 0.50 | -0.7% | -19.4% | 2.68 | 0.75 | 5.24 | 0.40 | +95.5% | -46.7% |

Table 6: Comparison of SOTA VLN Agents and Oracle (Ground-truth) on HA-VLN

| Method | Validation Seen (Diff.) | | | | Validation Unseen (Diff.) | | | |
|---|---|---|---|---|---|---|---|---|
| | NE ↓ | TCR ↓ | CR ↓ | SR ↑ | NE ↓ | TCR ↓ | CR ↓ | SR ↑ |
| Speaker-Follower [12] | +4.96 ↑ | +0.20 ↑ | +0.70 ↑ | −0.57 ↓ | +6.49 ↑ | +0.24 ↑ | +0.59 ↑ | −0.66 ↓ |
| Rec(Prevalent) [21] | +4.33 ↑ | +0.17 ↑ | +0.58 ↑ | −0.57 ↓ | +5.19 ↑ | +0.23 ↑ | +0.66 ↑ | −0.53 ↓ |
| Rec(OSCAR) [21] | +4.05 ↑ | +0.14 ↑ | +0.58 ↑ | −0.49 ↓ | +5.19 ↑ | +0.22 ↑ | +0.65 ↑ | −0.51 ↓ |
| Airbert [16] | +3.36 ↑ | +0.14 ↑ | +0.51 ↑ | -0.41 ↓ | +4.57 ↑ | +0.23 ↑ | +0.70 ↑ | -0.49 ↓ |
| Oracle | 0.62 | 0.04 | 0.175 | 0.91 | 0.67 | 0.008 | 0.037 | 0.89 |

both Navigation Error (NE) and Target Collision Rate (TCR), underscoring the critical importance of panoramic action spaces for effective and reliable navigation in complex, dynamically human-populated environments.

**Static vs. Dynamic Environment** (Tab. 3): Introducing dynamic human motion into the environment reduces SR by 46.7% in seen environments and by 19.4% in unseen settings, presenting a substantial obstacle to reliable and effective navigation while highlighting the challenges inherent in human-aware task performance.

Table 3: Static vs. Dynamic Environment Comparison

| Environment Type | Validation Seen | | Validation Unseen | |
|---|---|---|---|---|
| | NE ↓ | SR ↑ | NE ↓ | SR ↑ |
| Static | 2.68 | 0.75 | 4.01 | 0.62 |
| Dynamic | 5.24 | 0.40 | 3.98 | 0.50 |
| Difference | +2.56 | -0.35 | -0.03 | -0.12 |
| Percentage | +95.5% | -46.7% | -0.7% | -19.4% |

**Optimal vs. Sub-optimal Expert** (Tab. 2): Training with a sub-optimal expert marginally increases NE by 10.2% and reduces SR by 5.7% in seen environments. Although slightly lower in accuracy, sub-optimal expert guidance introduces greater realism to the agent's training, offering navigation experiences more aligned with real-world variability and thus contributing to improved robustness in human-aware metrics.

Table 7: Comparison of SOTA Agents on Traditional VLN vs. HA-VLN (Zero-shot)

| Method | Validation Seen | | | | | | Validation Unseen | | | | | |
|---|---|---|---|---|---|---|---|---|---|---|---|---|
| | w/o human | | w/ human | | Difference | | w/o human | | w/ human | | Difference | |
| | NE ↓ | SR ↑ | NE ↓ | SR ↑ | NE | SR | NE ↓ | SR ↑ | NE ↓ | SR ↑ | NE | SR |
| Speaker-Follower [12] | 6.62 | 0.35 | 7.12 | 0.24 | +7.6% | -31.4% | 3.36 | 0.66 | 4.96 | 0.40 | +47.6% | -39.4% |
| Rec (PREVALENT) [21] | 3.93 | 0.63 | 6.93 | 0.26 | +76.3% | -58.7% | 2.90 | 0.72 | 7.59 | 0.21 | +161.7% | -70.8% |
| Rec (OSCAR) [21] | 4.29 | 0.59 | 7.45 | 0.23 | +73.4% | -61.0% | 3.11 | 0.71 | 8.37 | 0.20 | +169.1% | -71.8% |
| Airbert [16] | 4.01 | 0.62 | 6.27 | 0.30 | +56.4% | -51.6% | 2.68 | 0.75 | 7.16 | 0.25 | +167.2% | -66.7% |

### 3.3 Evaluation of SOTA VLN Agents on the HA-VLN Task

We evaluated state-of-the-art (SOTA) Vision-and-Language Navigation (VLN) agents on the Human-Aware Vision-and-Language Navigation (HA-VLN) task. Each agent was adapted for HA-VLN by incorporating panoramic action spaces and sub-optimal expert guidance to navigate dynamic, human-occupied environments. Our evaluations included both retrained

Table 5: Human Perception on HA-VLN (Retrained)

| Method | Validation Seen | | Validation Unseen | |
|---|---|---|---|---|
| | TCR ↓ | CR ↓ | TCR ↓ | CR ↓ |
| Speaker-Follower [12] | 0.24 | 0.87 | 0.25 | 0.63 |
| Rec(Prevalent) [21] | 0.21 | 0.75 | 0.24 | 0.70 |
| Rec(OSCAR) [21] | 0.18 | 0.75 | 0.23 | 0.69 |
| Airbert [16] | 0.18 | 0.68 | 0.24 | 0.74 |

and zero-shot performance assessments, revealing substantial performance degradations in HA-VLN scenarios compared to traditional VLN tasks and significant gaps from the oracle, underscoring the increased complexity introduced by human-aware navigation.

**Retrained Performance.** In retrained HA-VLN settings, even the best-performing agent achieved a maximum success rate (SR) of only 40% in unseen environments, which is 49% lower than the oracle's SR (Tab. 4, Tab. 6). The impact of human occupancy is marked, with SR reductions of up to 65% in unseen settings. Despite retraining, agents remain limited in their human-aware capabilities, exhibiting high Target Collision Rates (TCR) and Collision Rates (CR). For instance, the Speaker-Follower model records TCR and CR values of 0.24 and 0.87 in seen environments, which contrast

sharply with the oracle's significantly lower TCR of 0.04 and CR of 0.175 (Tab. 5, Tab. 6). These disparities highlight the challenges agents face in adapting to human-centered dynamics.

**Zero-shot Performance.** The zero-shot performance of SOTA VLN agents in HA-VLN environments reveals even more pronounced challenges. While leading agents achieve up to 72% SR in traditional VLN tasks for unseen environments, this drops significantly under HA-VLN constraints (Tab. 7). Even Airbert, designed to manage complex environmental contexts, struggles in human-occupied settings, with navigation errors rising by over 167% and SR falling by nearly 67%. These results highlight the considerable difficulty agents encounter in dynamic, human-centric settings, emphasizing the necessity for further advancements in training strategies and navigation models to improve robustness and adaptability in real-world, human-aware navigation tasks.

Table 8: Performance Comparison of Our Proposed Agents on HA-VLN Tasks.

| Method | Proportion | Validation Seen | | | | Validation Unseen | | | |
|---|---|---|---|---|---|---|---|---|---|
| | | NE↓ | TCR↓ | CR↓ | SR↑ | NE↓ | TCR↓ | CR↓ | SR↑ |
| **VLN-DT (Ours)** | 100% | 8.51 | **0.30** | 0.77 | **0.21** | **8.22** | **0.37** | **0.58** | 0.11 |
| **VLN-CM (Ours)** | 0% | **7.31** | 0.38 | **0.73** | 0.19 | 8.22 | 0.42 | 0.62 | **0.12** |
| | 3% | 7.23 | 0.75 | 0.87 | 0.20 | 8.23 | 0.82 | 0.61 | 0.13 |
| | 25% | 7.85 | 0.85 | 0.61 | 0.16 | 8.42 | 0.99 | 0.52 | 0.12 |
| | 50% | 8.67 | 0.98 | 0.52 | 0.11 | 8.74 | 1.15 | 0.45 | 0.09 |
| | 100% | 10.61 | 1.01 | 0.62 | 0.03 | 10.39 | 1.14 | 0.48 | 0.02 |

## 3.4 Evaluation of Agents on HA-VLN Task

In this work, we introduce two agent models: the Vision-Language Navigation Decision Transformer (VLN-DT), trained on a dataset generated via random walk, and the Vision-Language Navigation Cross-Modal (VLN-CM), trained under expert supervision. This section compares their performance and examines the impact of various reward strategies on task execution.

**Performance Comparison.** Table 8 presents a comparative analysis of our agents on HA-VLN tasks. VLN-DT, trained with 100% random walk data, demonstrates comparable performance to the expert-supervised VLN-CM, exhibiting strong generalization capabilities. Notably, VLN-CM's performance degrades significantly as the proportion of random walk data increases; with 100% random data, Success Rate (SR) declines by 83.6% in seen and 81.5% in unseen environments. This outcome underscores VLN-DT's robustness and reduced dependency on expert guidance, making it well-suited for diverse and unpredictable scenarios.

**Reward Strategy Analysis.** Figure 5 illustrates the effect of different reward strategies on VLN-DT's performance. A straightforward reward for decreasing target distance resulted in inefficient trajectories with an elevated collision rate. Introducing a penalty-based distance reward achieved modest improvements in Success Rate (SR) and Collision Rate (CR). However, applying additional penalties for human collisions did not signif-

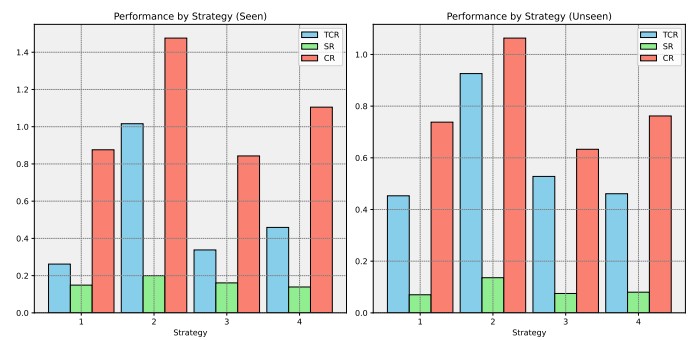

Figure 5: Effects of Reward Strategies on VLN-DT.

icantly enhance performance, underscoring the need for more advanced, human-aware reward strategies to effectively navigate agents through dynamic, human-populated environments.

This analysis highlights the advantages of VLN-DT's design in balancing adaptability and efficiency across various conditions while identifying key areas for future development in reward strategies tailored for human-aware navigation. Detailed performance metrics can be found in Appendix D.4.

## 3.5 Evaluation on Real-World Robots

To assess real-world applicability, we deployed our trained agent on a Unitree quadruped robot equipped with a stereo fisheye camera, ultrasonic distance sensors, and an inertial measurement unit

(IMU) (Fig. 15). The agent operates on an NVIDIA Jetson TX2, processing RGB images to make action inferences, which are subsequently executed via a Raspberry Pi 4B. Continuous IMU feedback enables the robot to monitor and adjust its movement for precision.

Experiments were conducted in office environments to evaluate the agent's navigation performance both in the absence and presence of humans. In human-free scenarios (Fig. 16), the agent successfully demonstrated accurate navigation by reliably following prescribed instructions. In human-populated settings, the agent exhibited human-aware navigation, detecting and actively avoiding individuals in its path (Fig. 17). However, we also observed cases where the robot's performance degraded, resulting in collisions due to sudden, unpredictable changes in human behavior (Fig. 18), which highlights the inherent challenges of navigating dynamic, human-centric environments.

These experiments underscore the effectiveness of transferring learned policies from simulated settings to physical robots, while also revealing areas for improvement. Specifically, the findings highlight the necessity for enhanced robustness and adaptability to better manage real-world complexity. Additional experimental details and results are provided in App. D.4.

## 4 Discussion

**Applications & Extensions.** The HA3D simulator advances the field of human-centered simulation by accommodating widely-adopted 3D formats, including .obj and .glb, thus streamlining integration and promoting broader research utility. This adaptability enables researchers to expand character diversity and customize agents within simulated scenes, fostering the creation of complex, multi-agent interactive environments. Moreover, the framework's architecture readily supports the incorporation of additional dynamic entities, such as animals and autonomous robots, thereby further enhancing the simulation's capacity to represent realistic, richly populated scenarios. For implementation details, please refer to our GitHub repository.

**Limitations.** While the Human-Aware Vision and Language Navigation (HA-VLN) framework constitutes a significant step forward in embodied AI navigation, certain limitations persist. The framework's current scope captures human presence and basic movement but does not yet model the breadth of human behavioral patterns and social nuances, which may affect the robustness of trained agents in real-world applications where human interactions are more complex and varied. Additionally, the HA3D and HA-R2R datasets are confined to indoor environments, which may limit the generalizability of trained agents across diverse real-world settings, particularly in outdoor contexts where navigation dynamics differ substantially.

**Future Work.** To further enhance the HA-VLN framework, future research should prioritize refining human behavior modeling to encompass more sophisticated social interactions, nuanced group dynamics, and contextualized interpersonal behaviors. The inclusion of avatars with heightened behavioral fidelity would enrich the simulation's realism, enabling more effective modeling of human-agent interactions. Extending the simulator to support outdoor environments is also paramount, as this expansion would allow for the development of agents capable of navigating across a wider range of real-world scenarios. These improvements, coupled with advanced domain adaptation techniques and robust strategies for managing environmental uncertainty, are essential to foster the development of highly adaptable and resilient VLN systems capable of seamless operation within diverse, human-populated environments.

## 5 Conclusion

This work presents the Human-Aware Vision and Language Navigation (HA-VLN) framework, which integrates dynamic human activities while relaxing restrictive assumptions inherent to conventional VLN systems. Through the development of the Human-Aware 3D (HA3D) simulator and the Human-Aware Room-to-Room (HA-R2R) dataset, we provide a comprehensive environment for the training and evaluation of HA-VLN agents. We introduce two agent architectures—the Expert-Supervised Cross-Modal (VLN-CM) and the Non-Expert-Supervised Decision Transformer (VLN-DT)—each leveraging cross-modal fusion and diverse training paradigms to support effective navigation in dynamically populated settings. Extensive evaluation highlights the contributions of this framework while underscoring the need for continued research to strengthen HA-VLN agents' robustness and adaptability for deployment in complex, real-world environments.

## Author Contributions

Heng Li was responsible for agent development and evaluations, drafted the initial agent and evaluation sections, and revised the final manuscript based on review feedback. Minghan Li was responsible for simulator development and evaluations, prepared the initial draft of the simulator and evaluation sections, conducted real-world testing, and created the project website. Zhi-Qi Cheng supervised the design and development of both the agent and simulator, managed the entire project execution, designed the evaluation plan, drafted the initial manuscript, revised the final version, and provided guidance to the entire team. Yifei Dong designed the initial simulation prototyping, drafted the related work section, and provided revision suggestions. Yuxuan Zhou offered collaborative feedback and contributed revision suggestions. Jun-Yan He participated in project discussions. Qi Dai provided invaluable strategic guidance and contributed to manuscript revisions. Teruko Mitamura offered constructive feedback, and Alexander G. Hauptmann provided critical insights and contributed to manuscript refinement. We also thank the anonymous reviewers for their valuable suggestions.

## Acknowledgments

This work was partially supported by the Air Force Research Laboratory under agreement number FA8750-19-2-0200; the financial assistance award 60NANB17D156 from the U.S. Department of Commerce, National Institute of Standards and Technology (NIST); the Intelligence Advanced Research Projects Activity (IARPA) via the Department of Interior/Interior Business Center (DOI/IBC) contract number D17PC00340; and the Defense Advanced Research Projects Agency (DARPA) grant under the GAILA program (award HR00111990063) and the AIDA program (award FA8750-18-20018). Additional support was provided by the Carnegie Mellon Manufacturing Futures Institute and the Manufacturing PA Innovation Program.

The authors also acknowledge the Intel Ph.D. Fellowship and the IBM Outstanding Students Scholarship awarded to Zhi-Qi Cheng, as well as the computing resources provided by Microsoft Research. We further extend our gratitude to the School of Computer Science (SCS) at Carnegie Mellon University, particularly the High Performance Computing (HPC) facility, for providing essential computational resources.

The U.S. Government is authorized to reproduce and distribute reprints for governmental purposes notwithstanding any copyright notation therein. The views and conclusions presented in this work are those of the authors and do not necessarily represent the official policies or endorsements, either expressed or implied, of the Air Force Research Laboratory, NIST, IARPA, DARPA, Microsoft Research, or other funding agencies.

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

# A  Related Work

We trace the evolution of the Visual-and-Language Navigation (VLN) task and highlight the key differences between our proposed Human-Aware VLN (HA-VLN) task and prior work, focusing on three critical aspects: *Egocentric Action Space*, *Human Interactivity*, and *Sub-optimal Expert*. Tab. 9 provides a detailed comparison of tasks, simulators, and agents based on these aspects.

**Evolution of VLN Tasks**. VLN originated with tasks like Room-to-Room (R2R) [2, 15, 27] for indoor navigation, while TOUCHDOWN and MARCO [7, 34] focused on outdoor navigation. Goal-driven navigation with simple instructions was explored in REVERIE[40] and VNLA[37], and DialFRED[13] and CVDN[45] introduced navigation through human dialogue. However, since the Speaker-Follower [12], panoramic action spaces have been predominantly used, deviating from our first assumption of an Egocentric Action Space, which provides a more realistic and challenging navigation scenario. More recent tasks, such as Room-for-Room (R4R), RoomXRoom, VNLA, CVDN, and VLN-CE[22, 26, 27, 37, 45], have started to address dynamic navigation scenarios in Egocentric Action Space. Nevertheless, they still lack the complexity of real-world human interactions that HA-VLN specifically targets, which is crucial for developing agents that can navigate effectively in the presence of humans.

**Simulator for VLN Tasks**. VLN simulators can be categorized into photorealistic and non-photorealistic. Non-photorealistic simulators like AI2-THOR[25] and Gibson GANI [50] do not include human activities, while photorealistic simulators such as House3D [49], Matterport3D [2], and Habitat [42] offer high visual fidelity but typically lack dynamic human elements. The absence of human interactivity in these simulators limits their ability to represent real-world navigation scenarios, which is crucial for our second assumption of Human Interactivity. Some simulators, like Habitat3.0[39], AI2-THOR[25], and ViZDoom[24], consider human interaction but provide non-photorealistic scenes, while Google Street View offers a photorealistic outdoor environment with static humans. In contrast, our HA3D simulator bridges the gap between simulated tasks and real-world applicability by integrating photorealistic indoor environments enriched with human activities, enabling the development of agents that can navigate effectively in the presence of dynamic human elements.

**Agent for VLN Tasks**. Early VLN models, enhanced by attention mechanisms and reinforcement learning algorithms [13, 33, 40, 47], paved the way for recent works based on pre-trained visual-language models like ViLBert [32]. These models, such as VLN-BERT[36], PREVALENT[18], Oscar[29], Lily[30], and ScaleVLN[48], have significantly improved navigation success rates by expanding the scale of pre-training data. However, most of these agents navigate using a panoramic action space, unlike [2, 54, 55], which operate in an Egocentric action space. Notably, NaVid[54] demonstrated the transfer of the agent to real robots. Despite these advancements, most of these agents are guided by an optimal expert, which conflicts with our third assumption of using a sub-optimal expert. In real-world scenarios, expert guidance may not always be perfect, and agents need to be robust to handle such situations. Our agents are specifically designed to operate effectively under less stringent and more realistic expert supervision, enhancing their ability to perform in true Sim2Real scenarios and setting them apart from previous approaches.

Table 9: Comparison of Tasks, Simulators, and Agents based on the three key aspects: Egocentric Action Space, Human Interactivity, and Sub-optimal Expert.

| | Egocentric Action Space | Human | Sub-optimal Expert | Previous Work |
|---|---|---|---|---|
| **Tasks** | – | ✗ | – | EQA [10], IQA [14] |
| | ✗ | ✗ | ✗ | MARCO [34], DRIF [4], VLN-R2R [2], TOUCHDOWN [7], REVERIE [40], DialFRED [13] |
| | ✓ | ✗ | ✗ | VNLA [37], CVDN [45], VLN-CE [26], Room4Room[22], RoomXRoom[27] |
| | ✓ | ✓ | ✓ | HA-VLN(Our) |
| **Simulators** | – | ✗ | – | Matterport3D[2], House3D[49], AI2-THOR [25], Gibson GANI [50], Habitat [42] |
| | – | ✓ | – | HA-VLN(Our), Habitat3.0 [39], Google Street, ViZDoom [24] |
| **Agents** | ✗ | ✗ | ✗ | EnvDrop [44], AuxRN [56], PREVALENT [18], RelGraph [20], HAMT [9] Rec-VLNBERT[21], EnvEdit[28], Airbert [16], Lily [30], ScaleVLN [48] |
| | ✓ | ✗ | ✗ | NavGPT [55], NaVid [54], Student Force [2] |
| | ✓ | ✓ | ✓ | HA-VLN Agent |

## B  Simulator Details

The HA3D simulator's code structure is inspired by the Matterport3D (MP3D) simulator, which can be found at https://github.com/peteanderson80/Matterport3DSimulator. To obtain access to the Matterport Dataset, we sent an email request to matterport3d@googlegroups.com. The source code for the HA3D simulator is available in our GitHub repository at https://github.com/lpercc/HA3D_simulator. As illustrated in Fig. 6, the HA3D simulator provides agents with three key features that distinguish it from traditional VLN frameworks: an Ergonomic Action Space, Dynamic Environments, and a Sub-Optimal Expert.

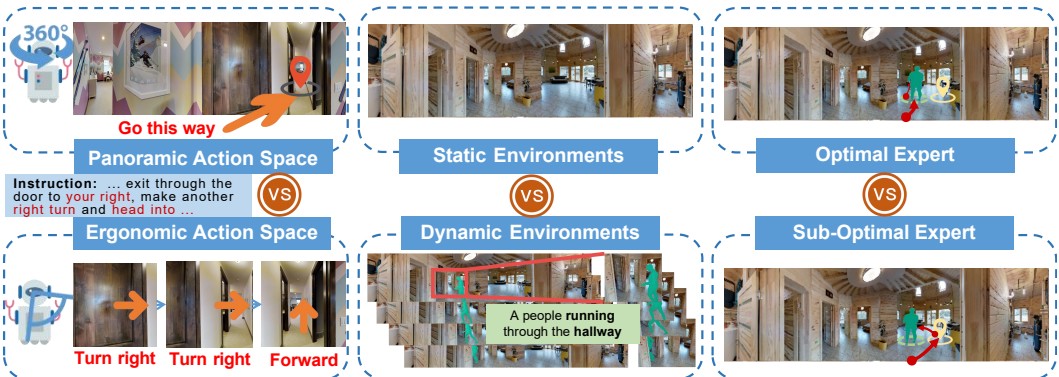

Figure 6: Overview of the VLN framework assumptions in the HA3D simulator. The simulator introduces an Ergonomic Action Space, Dynamic Environments, and a Sub-Optimal Expert to bridge the gap between simulated and real-world navigation scenarios. The Ergonomic Action Space limits the agent's field of view to 60 degrees, requiring a more realistic navigation strategy compared to the panoramic view used in traditional VLN tasks. Dynamic Environments incorporate time-varying elements, such as human activities, challenging the agent to adapt its navigation strategy to handle video streams that include people. The Sub-Optimal Expert provides navigation guidance that accounts for human factors and dynamic elements, resulting in a more realistic and human-like navigation strategy compared to the optimal expert model that always finds the shortest path without considering these factors. [Best viewed in color]

### B.1  HAPS Dataset

The HAPS Dataset encompasses a diverse range of 29 indoor regions, including *bathroom, bedroom, closet, dining room, entryway/foyer/lobby, family room, garage, hallway, library, laundry room/mudroom, kitchen, living room, meeting room/conference room, lounge, office, porch/terrace/deck/driveway, recreation/game room, stairs, toilet, utility room/tool room, TV room, workout/gym/exercise room, outdoor areas containing grass, plants, bushes, trees, etc., balcony, other room, bar, classroom, dining booth, and spa/sauna*. The dataset features skinned human motion models devoid of identifiable biometric features or offensive content. Fig. 9 illustrates the skeletons of the dataset's human activities, accompanied by their corresponding descriptions, which exhibit diverse forms and interactions with the environment.

To ensure the quality and relevance of the human activity descriptions, we employed GPT-4 to generate an extensive set of descriptions for each of the 29 indoor regions. Subsequently, we conducted a rigorous human survey involving 50 participants from diverse demographics to evaluate and select the most appropriate descriptions. As depicted in Fig. 7, each participant assessed the descriptions for a specific indoor region based on three key criteria: 1) High Relevance to the specified region, 2) Verb-Rich Interaction with the environment, and 3) Conformity to Daily Life patterns.

The survey was conducted in five rounds, with the highest-rated descriptions from previous rounds being excluded from subsequent evaluations to ensure a comprehensive review process. Upon analyzing the survey responses, we identified the activity descriptions with the highest selection frequency for each region, ultimately curating a set of 145 human activity descriptions (Fig. 8).

The resulting HAPS Dataset, available for download at https://drive.google.com/drive/folders/1aswHATnKNViqw6QenAwdQRTwXQQE5jd3?usp=sharing, represents a meticulously crafted resource for studying and simulating human activities in indoor environments.

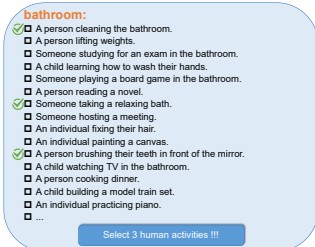

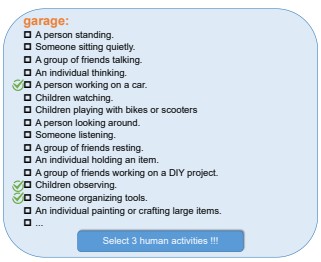

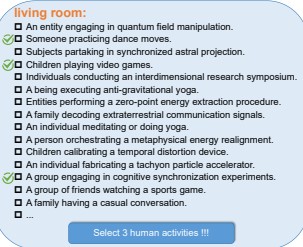

(a) Criterion 1: High relevance between human activities and their respective regions

(b) Criterion 2: Human activities contain verb-rich interactions with the environment

(c) Criterion 3: Human activities conform to everyday life patterns and colloquial language

Figure 7: Criteria for filtering suitable human activity descriptions through human surveys. The three key criteria ensure the relevance, interactivity, and realism of the selected activities, resulting in a curated set of 145 human activity descriptions for the HAPS Dataset. [Zoom in to view]

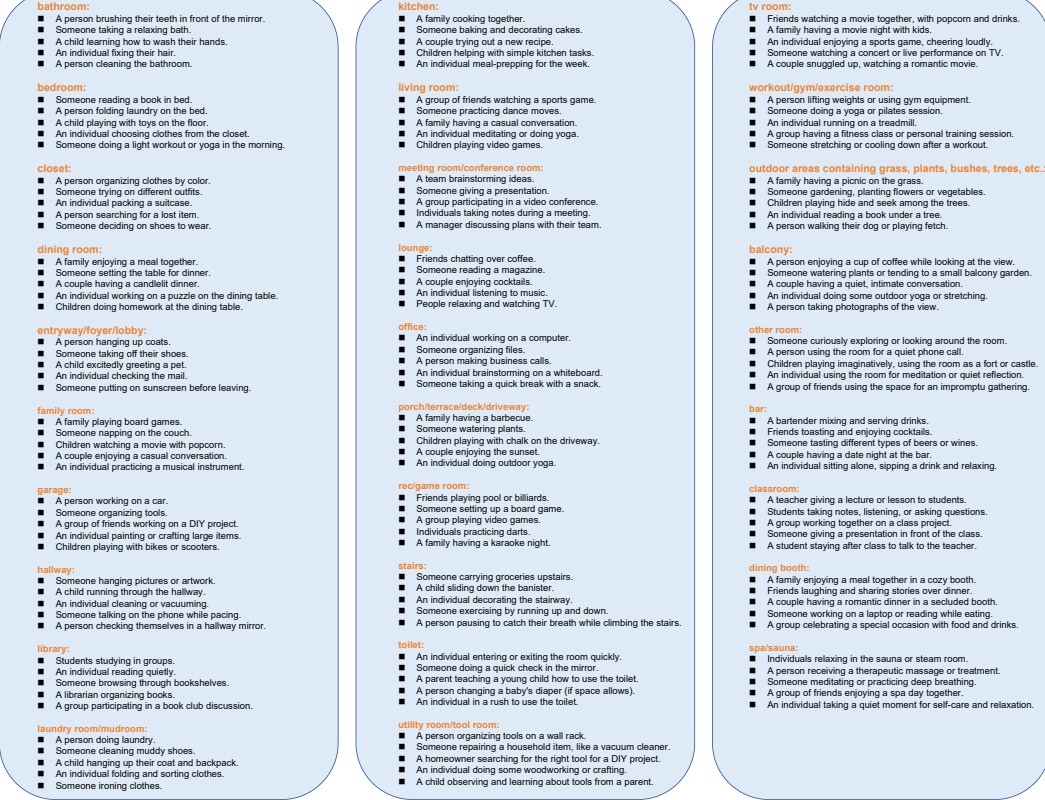

Figure 8: The 145 human activity descriptions in the HAPS Dataset, categorized by their respective indoor regions (highlighted in bold red font). Each region includes 5 carefully selected human activity descriptions that best represent the diversity and relevance of activities within that space. [Zoom in to view]

## B.2 Human Activity Annotation

To facilitate a comprehensive understanding of the HA-VLN task environment, we present a large-scale embodied agents environment with the following key statistical insights:

**Human Distribution by Region.** As illustrated in Fig. 10(a), a total of 374 humans are distributed across the environment, with an average of four humans per building. This distribution ensures a realistic and dynamic navigation setting, closely mimicking real-world scenarios.

**Human Activity Trajectory Lengths.** Fig. 10(b&c) showcases the distribution of human activity trajectory lengths. The total trajectory length spans 1066.81m, with an average of 2.85m per human. Notably, 49.2% of humans engage in stationary activities (less than 1 meter), 30.5% move short

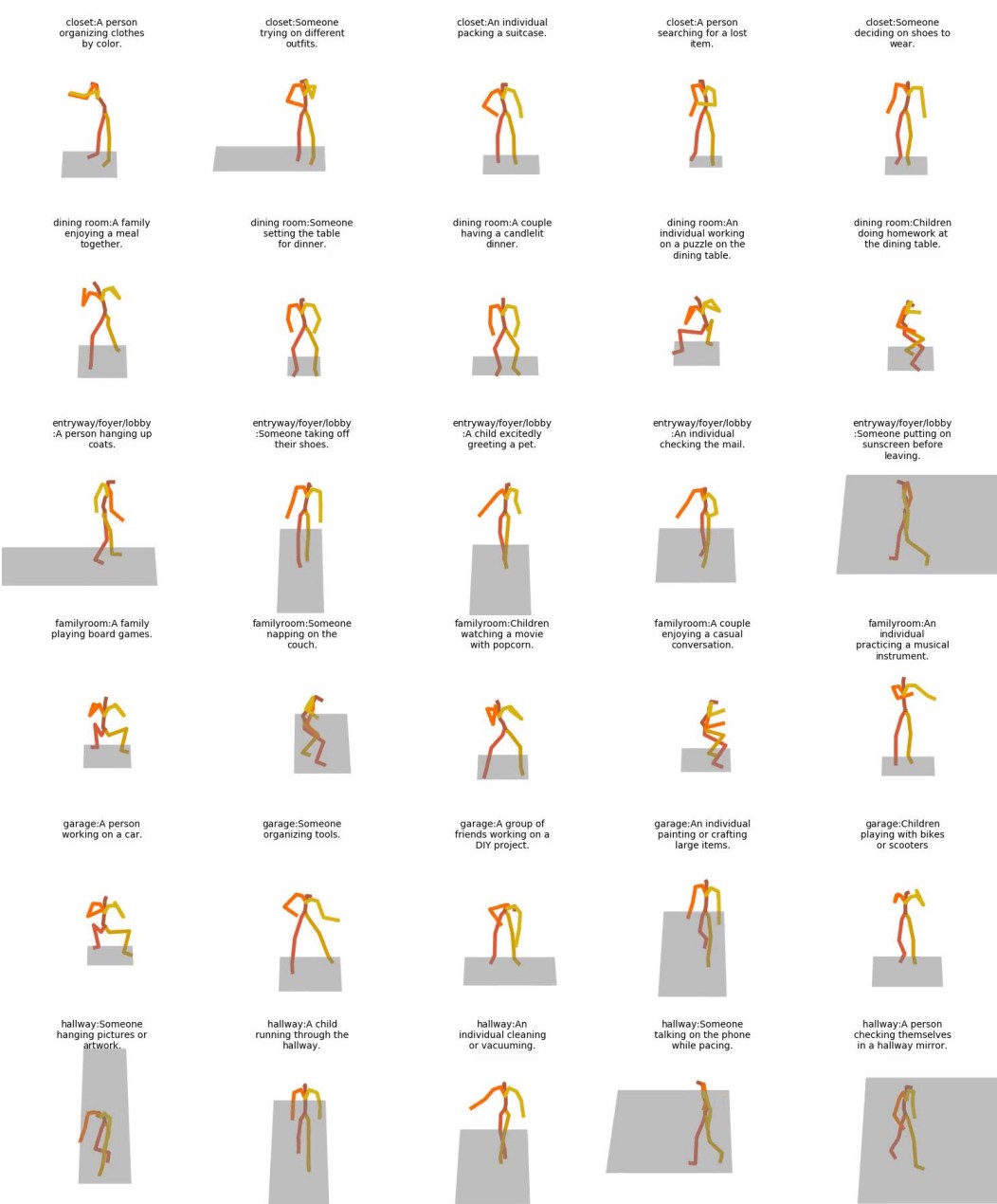

Figure 9: Human skeletons in the HAPS Dataset, showcasing the diversity of human activities across 6 common indoor regions. Each row represents five different activity descriptions within the same indoor region, with the corresponding activity description displayed above each human skeleton diagram. The HAPS Dataset captures a wide range of realistic and interactive human behaviors. [Zoom in to view]

distances (1-5 meters), 18.4% move long distances (5-15 meters), and 1.9% move very long distances (more than 15 meters). This diverse range of trajectory lengths captures the varied nature of human activities within indoor environments.

**Human Impact on the Environment.** The presence of humans significantly influences the navigation environment, as depicted in Fig. 10(d). Among the 10,567 viewpoints in the environment, 8.16% are directly affected by human activities, i.e., viewpoints through which humans pass. Furthermore, 46.47% of the viewpoints are indirectly affected, meaning that humans are visible from these locations. This substantial impact highlights the importance of considering human presence and movement when developing navigation agents for real-world applications.

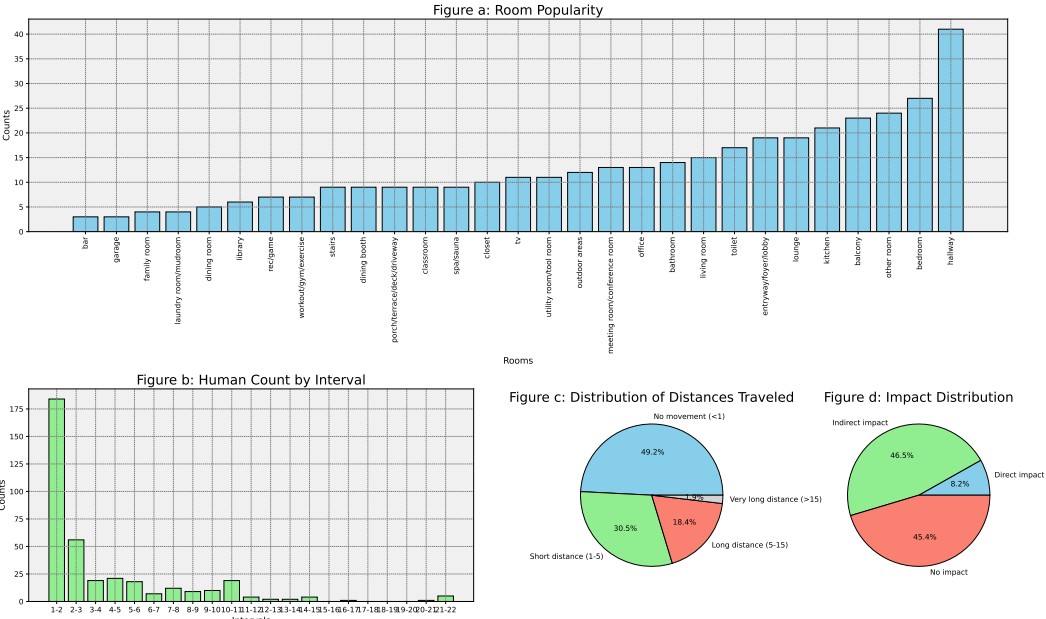

Figure 10: Statistics on human distribution in the HA-VLN environment. (a) Distribution of humans by region, showcasing the average number of humans per building. (b) Distribution of human activity trajectory lengths, categorized by stationary, short, long, and very long distances. (c) Percentage breakdown of human activity trajectory lengths. (d) Impact of human presence on the environment, illustrating the percentage of viewpoints directly and indirectly affected by human activities. [Zoom in to view]

## B.3 Realistic Human Rendering

The rendering process has been meticulously optimized to ensure spatial and visual coherence between human motion models and the scene. Fig. 11 showcases the realistic rendering of humans in various indoor environments, demonstrating the simulator's ability to generate lifelike and visually diverse scenarios. The following key optimizations contribute to high-quality rendering:

**Camera Alignment with Agent's Perspective.** The rendering process aligns the camera settings with the agent's perspective, incorporating a 60-degree field of view (FOV), 120 frames per second (fps), and a resolution of 640x480 pixels. This alignment ensures that the rendered visuals accurately mirror the agent's visual acuity and motion fluidity, providing a realistic and immersive experience.

**Integration of Human Motion Models.** To generate continuous and lifelike movements, the simulator leverages 120-frame sequences of SMPL mesh data when placing human motion models in the scene. This approach allows for the sequential output of both RGB and depth frames, effectively capturing the dynamics of human motion and enhancing the realism of the rendered environment.

**Utilization of Depth Maps.** The rendering process employs depth maps to distinctly segregate the foreground (*human models*) from the background (*scene*). By doing so, the simulator ensures that the rendered humans accurately integrate with the environmental context without visual discrepancies, resulting in a seamless and visually coherent experience. Fig. 13 presents continuous video frames captured from the HA3D simulator. These optimizations ensure that the HA3D simulator provides a high level of realism and detail in rendering human activities within indoor environments. By accurately replicating human movements and interactions, the simulator creates a rich and dynamic setting for training and evaluating human-aware navigation agents.

These optimizations ensure that the HA3D simulator provides a high level of realism and detail in rendering human activities within indoor environments. By accurately replicating human movements and interactions, the simulator creates a rich and dynamic setting for training and evaluating human-aware navigation agents. By incorporating adjustable video observations, navigable viewpoints, and collision feedback signals, the HA3D simulator offers a comprehensive and flexible environment for advancing research in human-aware vision-and-language navigation. These features ensure that

the agents developed and tested within this simulator are well-prepared for the complexities and challenges of real-world navigation tasks.

## B.4    Agent-Environment Interaction

To ensure the versatility and applicability of the HA3D simulator across a wide range of navigation tasks, we have designed the agent's posture and basic actions to align with the configurations of the well-established Matterport3D simulator. This design choice facilitates a seamless transition for researchers and practitioners, allowing them to leverage their existing knowledge and methodologies when utilizing the HA3D simulator. At each time step, agents within the HA3D simulator can receive several critical environmental feedback signals that enhance their understanding of the dynamic navigation environment.

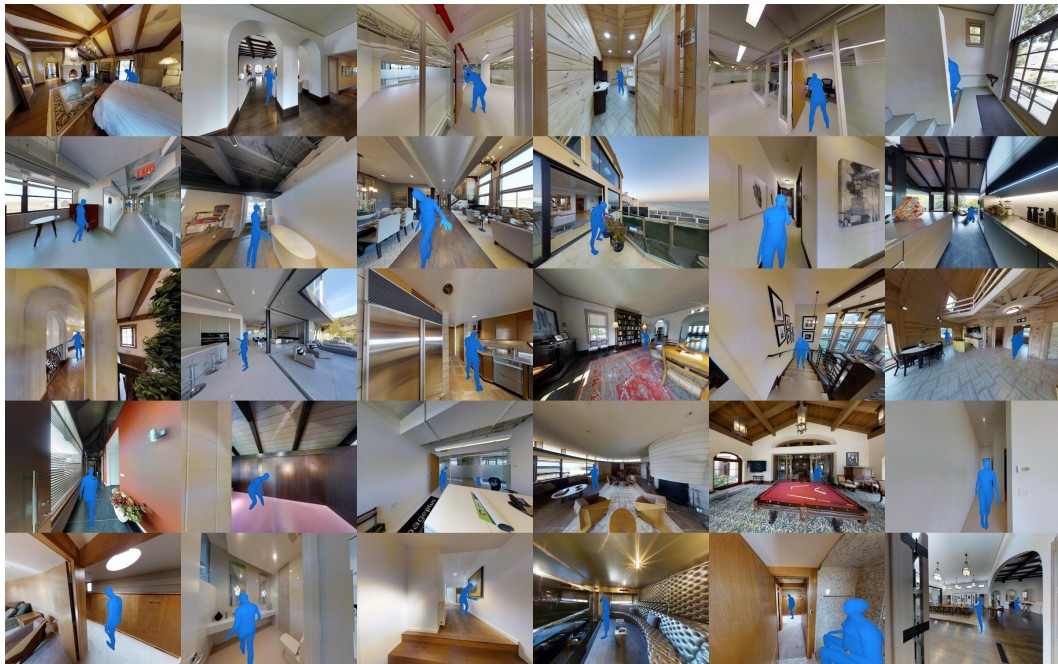

Figure 11: Single-frame in the HA3D simulator showcase viewpoints with human presence in each scene(120-degree FOV), demonstrating the diversity of human activities and environments. Common indoor regions such as bedrooms, hallways, kitchens, balconies, and bathrooms are displayed. Multiple humans can appear in the same region, as seen in the third row, sixth column, and the fifth row, fifth and sixth columns. [Zoom in to view]

**Set of Navigable Viewpoints.** The HA3D simulator provides agents with reachable viewpoints around them, referred to as navigable viewpoints. This feature enhances the navigation flexibility and practicality of the simulator, allowing agents to make informed decisions based on their current position and the available paths. By providing agents with a set of navigable viewpoints, the simulator empowers them to explore the environment efficiently and effectively, mimicking the decision-making process of real-world navigational agents.

**Human "Collision" Feedback Signal.** To promote safe and socially-aware navigation, the HA3D simulator incorporates a human "collision" feedback signal. Specifically, when the distance between an agent and a human falls below a predefined threshold (default: 1 meter), the simulator triggers a feedback signal, indicating that the human has been "crushed" by the agent. This feedback mechanism serves as a critical safety measure, encouraging agents to maintain a safe distance from humans and avoid potential collisions. By integrating this feedback signal, the simulator reinforces the importance of socially-aware navigation and facilitates the development of algorithms that prioritize human safety in dynamic environments.

## B.5    Implementation and Performance

The HA3D Simulator is a powerful and efficient platform designed specifically for simulating human-aware navigation scenarios. Built using a combination of C++, Python, OpenGL, and Pyrender,

the simulator seamlessly integrates with popular deep learning frameworks, enabling researchers to efficiently train and evaluate navigation agents in dynamic, human-populated environments. One of the key strengths of the HA3D Simulator is its customizable settings, which allow researchers to tailor the environment to their specific requirements. Users can easily adjust parameters such as image resolution, field of view, and frame rate, ensuring that the simulator can accommodate a wide range of research objectives and computational constraints. In terms of performance, the HA3D Simulator

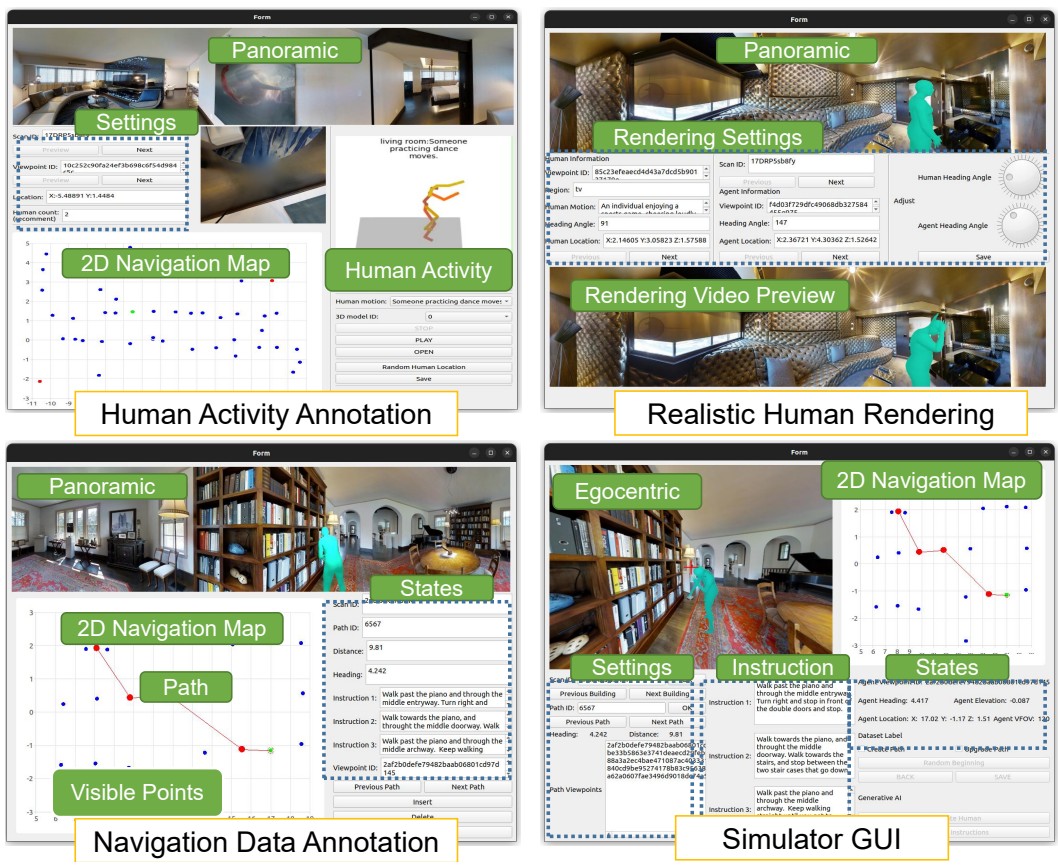

Figure 12: HA3D simulator interfaces and components, showcasing adjustments to human actions and activities. The interactive annotation tool enables users to locate humans in different building regions, set initial positions, and select 3D human motion models. [Zoom in to view]

achieves impressive results, even on modest hardware. When running on an NVIDIA RTX 3050 GPU, the simulator can maintain a frame rate of up to 300 fps at a resolution of 640x480. This level of performance is comparable to state-of-the-art simulation platforms [49, 39, 42], demonstrating the simulator's efficiency and optimization. Resource efficiency is another notable aspect of the HA3D Simulator. On a Linux operating system, the simulator boasts a memory footprint of only 40MB, making it accessible to a wide range of computing environments. Additionally, the simulator supports multi-processing operations, enabling researchers to leverage parallel computing capabilities and significantly enhance training efficiency.

To further facilitate the annotation process and improve accessibility, we have developed a user-friendly annotation toolset based on PyQt5 (Fig. 12). These tools feature an intuitive graphical user interface (GUI) that allows users to efficiently annotate human viewpoint pairs, motion models, and navigation data. The annotation toolset streamlines the process of creating rich, annotated datasets for human-aware navigation research.

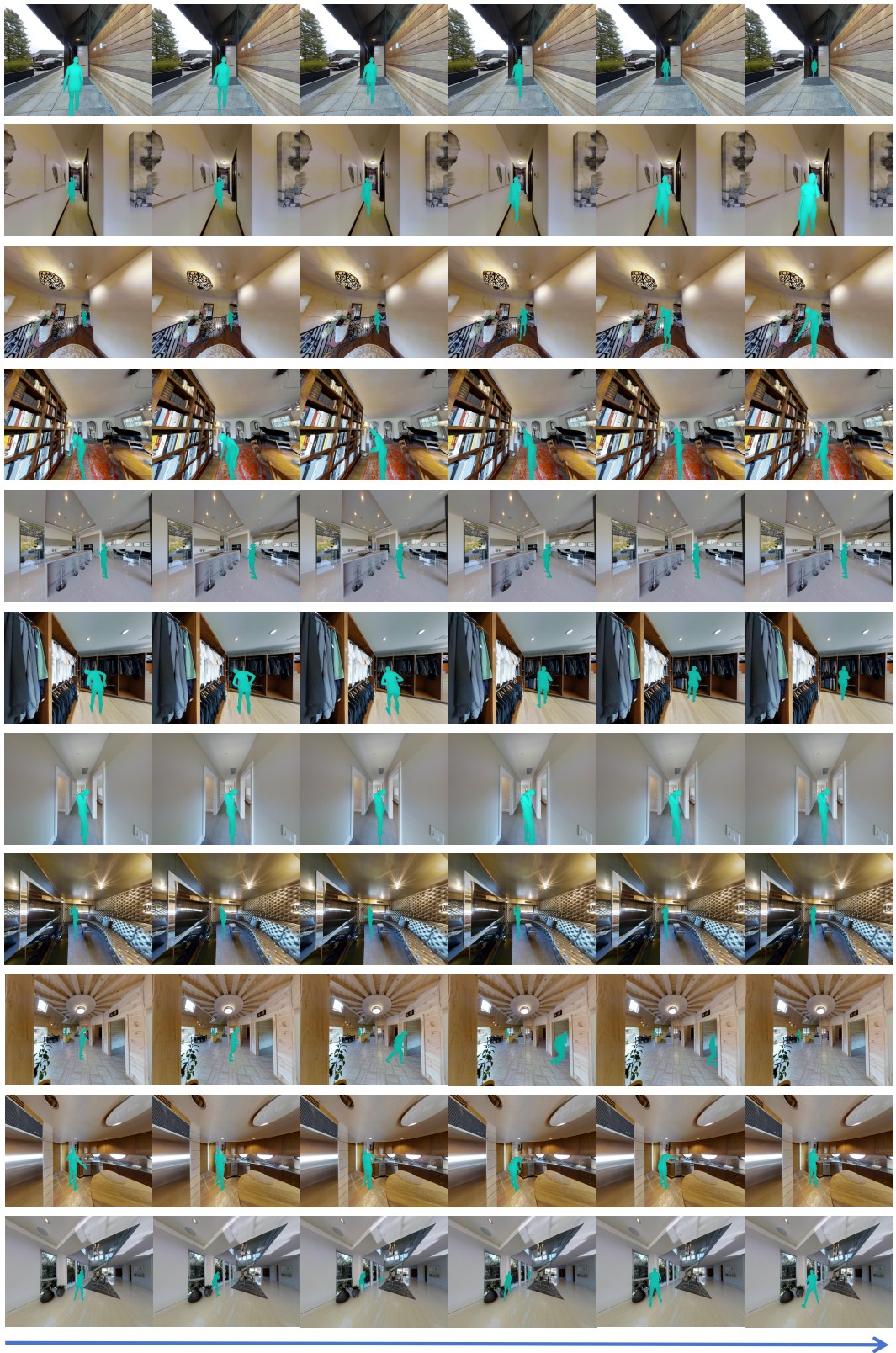

Figure 13: Video frames in the HA3D simulator showcasing viewpoints with human presence in each scene (120-degree FOV), reflecting visual diversity. Common indoor regions such as hallways, offices, dining rooms, closets, TV rooms, living rooms, and bedrooms are displayed. The simulator is capable of rendering multiple humans within the same region and field of view, as exemplified in the 9th and 11th rows of the grid, where two people appear simultaneously. [Zoom in to view]

# C Agent Details

## C.1 HA-R2R Dataset

**Instruction Generation.** To generate new instructions for the HA-R2R dataset, we utilize `LangChain` and `sqlang` to interface with `GPT-4`, leveraging its powerful language generation capabilities to create contextually relevant and coherent instructions. Note that we use `GPT-4 Turbo` in our code; it refers to the model ID `gpt-4-1106-preview` in the OpenAI API. Our approach to instruction generation involves the use of a carefully designed few-shot template prompt. This prompt serves as a guiding framework for the language model, providing it with the necessary context and structure to generate instructions that align with the objectives of the HA-R2R dataset.

The few-shot template prompt consists of two key components: a system prompt and a set of few-shot examples. The system prompt is designed to prime the language model with the overall context and requirements for generating navigation instructions in the presence of human activities. It outlines the desired characteristics of the generated instructions, such as their relevance to the navigation task, incorporation of human activity descriptions, and adherence to a specific format. The few-shot examples, on the other hand, serve as a sequence of representative instructions that demonstrate the desired output format and content. These examples are carefully curated to showcase the inclusion of human activity descriptions, the use of relative position information, and the integration of these elements with the original navigation instructions from the R2R dataset.

By providing both the system prompt and the few-shot examples, we effectively guide the generation process towards producing instructions that are consistent with the objectives of the HA-R2R dataset. List. 1 and List. 2 provide a detailed illustration of our prompt engineering approach, showcasing the system prompt and the few-shot examples used for sequential instruction generation. Through this prompt engineering technique, we are able to harness the power of `GPT-4` to generate a diverse set of new instructions that effectively incorporate human activity descriptions and relative position information, enhancing the realism and complexity of the navigation scenarios in the HA-R2R dataset.

```
Your role is to function as an
instruction generator. You will
receive Route-to-Route (R2R)
navigation instructions and information
about human activities.
Your task is to integrate these R2R
instructions with the details of
the human activities to create new,
clear instructions that a robot
can understand for navigation.

Please adhere to the following
guidelines when generating
instructions:

- Deliver the instructions in a single
paragraph
... // five rule to generation
- Conclude with a relative position
description, which should
be limited to one or two sentences.

Now I give you 5 examples.

... // examples in listing 2

Original instruction: ...
Human Activity Description: ...
Relative Position Description: ...

question: What is the instructure?
... \\ generation
```

Listing 1: Format of our LLM prompt

```
#Example 1

"original instruction":
"Walk between the columns and make a
    sharp turn right. Walk down the
    steps and stop on the landing. "

"human description":
"The 1th Human Description: Relative
    Position:Beginning Location and
    Activity: balcony:A couple having a
     quiet, intimate conversation."

"question" :
"What is the new instruction?"

"answer" :
"Navigate between the columns and
    execute a sharp right turn, taking
    special care at the beginning of
    your path where a couple might be
    engaged in a quiet, intimate
    conversation on the balcony.
    Proceed to walk down the steps and
    come to a halt on the landing,
    ensuring you do not disturb the
    couple's privacy or space during
    your movement."
```

Listing 2: Few-shot examples in our prompt

**Word Frequency Analysis.** To assess the quality and practicality of the instructions in the HA-R2R dataset, we conducted a comprehensive word frequency analysis. Fig. 14 shows the dataset's potential

to support the development and evaluation of robust navigation agents that can effectively interpret and follow human-like instructions in complex, dynamic environments.

The left chart in Fig. 14 illustrates the frequency of various nouns used in the instructions. The top 5 most frequent nouns are *turn*, *stair*, *room*, *hallway*, and *door*. Among these, the noun *turn* exhibits the highest frequency, appearing more than 5000 times throughout the dataset. Other nouns in the list include *exit*, *left*, *bedroom*, *right*, *bathroom*, *walk*, *doorway*, *towards*, *table*, *kitchen*, *area*, *way*, *step*, *proceed*, *chair*, *hall*, *bed*, *side*, *path*, and *living*. The presence of these nouns indicates the rich spatial and contextual information conveyed in the navigation instructions.

Similarly, the right chart in Fig. 14 presents the frequency distribution of various verbs used in the instructions. The top 5 most frequent verbs are *proceed*, *make*, *walk*, *turn*, and *leave*. Among these, the verb *proceed* exhibits the highest frequency, also appearing over 5000 times throughout the dataset. Other verbs in the list include *reach*, *take*, *continue*, *go*, *enter*, *begin*, *exit*, *stop*, *pass*, *keep*, *navigate*, *move*, *ascend*, *approach*, *descend*, *straight*, *ensure*, *be*, *follow*, and *locate*. The diversity of these verbs highlights the range of actions and directions provided in the navigation instructions.

The word frequency analysis provides valuable insights into the composition and quality of the HA-R2R dataset. The prevalence of common navigation instruction words, such as spatial nouns and action verbs, demonstrates the dataset's adherence to established conventions in navigation instruction formulation. This consistency ensures that the instructions are practical, easily understandable, and aligned with real-world navigation scenarios. Moreover, the balanced distribution of nouns and verbs across the dataset indicates the presence of rich spatial and temporal information in the instructions. The nouns provide crucial details about the environment, landmarks, and objects, while the verbs convey the necessary actions and movements required for successful navigation.

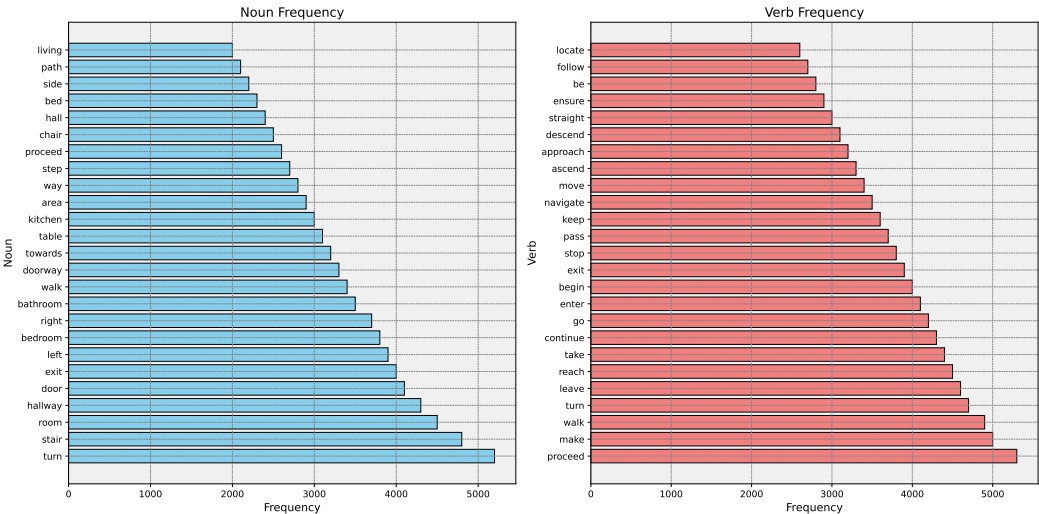

Figure 14: Word frequency distribution of all instructions in the HA-R2R dataset, showcasing the prevalence of common navigation instruction words. The x-axis of both charts represents the frequency range, while the y-axis lists the words. The bars are colored light blue for nouns (left chart) and light red for verbs (right chart), providing a clear visual distinction between the two word categories. The balanced distribution of nouns and verbs highlights the rich spatial and temporal information conveyed in the navigation instructions, ensuring their quality and practicality. [Zoom in to view]

## C.2 Algorithm to Construct Oracle(Expert) Agent

The Expert agent, also known as the Oracle agent, is handcrafted using a sophisticated path planning and collision avoidance strategy. The algorithm employed to construct the expert agent is summarized in algorithm 1. The Oracle agent operates by parsing the provided language instructions $\mathcal{I} = \langle w_1, w_2, \cdots, w_L \rangle$ and identifying the current state $\mathbf{s}_t = \langle \mathbf{p}_t, \phi_t, \lambda_t, \Theta_t^{60} \rangle$. It then updates the navigation graph $G = (N, E)$ by excluding the subset of nodes $N_h$ that are affected by human activity, resulting in a modified graph $G' = (N \setminus N_h, E')$. This step ensures that the agent avoids navigating through areas where human activities are present. Using the updated graph $G'$, the Oracle agent computes the shortest path to the goal using the A* search algorithm. This algorithm efficiently explores the navigation graph, considering the cost of each node and the estimated distance to the goal, to determine the optimal path.

If human activity is detected along the planned path, the Oracle agent employs a two-step approach for collision avoidance. First, it attempts to make a dynamic adjustment to its trajectory. If a safe alternative path is available, the agent selects the next state $\mathbf{s}_2'$ that minimizes the cost function $c(\mathbf{s}_2, \mathbf{s})$ while avoiding the human-occupied state $h_2$. This dynamic adjustment allows the agent to smoothly navigate around human activities without significantly deviating from its original path. In cases where dynamic adjustment is not possible, the Oracle agent resorts to rerouting. If the distance between the current state $\mathbf{s}_t$ and the human-occupied state $h_t$ is less than the avoidance threshold distance $\delta$, the agent reroutes to an alternative state $\mathbf{s}_t'$. This rerouting strategy ensures that the agent maintains a safe distance from human activities and prevents potential collisions. Throughout the navigation process, the Oracle agent continuously monitors the distance between its current state $\mathbf{s}_t$ and any human-occupied states $h_t$. If the distance falls below the minimum safe distance $\epsilon$, the collision indicator $\mathcal{C}(\mathbf{s}_t, h_t)$ is set to 1, signifying a potential collision. This information is used to guide the agent's decision-making and ensure safe navigation.

Finally, the Oracle agent executes the determined action $a_t$ and continues to navigate towards the goal until it is reached. By iteratively parsing instructions, updating the navigation graph, computing optimal paths, and employing dynamic adjustments and rerouting strategies, the Oracle agent effectively navigates through the environment while avoiding human activities and maintaining a safe distance.

---

**Algorithm 1** Oracle Agent Path Planning and Collision Avoidance Strategies

---

**Require:** Language instructions $\mathcal{I} = \langle w_1, w_2, \ldots, w_L \rangle$, current state $\mathbf{s}_t = \langle \mathbf{p}_t, \phi_t, \lambda_t, \Theta_t^{60} \rangle$, navigation graph $G = (N, E)$, subset of nodes affected by human activity $N_h$, minimum safe distance $\epsilon$, avoidance threshold distance $\delta$
**Ensure:** Next action $a_t$
  **while** goal not reached **do**
    Parse $\mathcal{I}$, identify $\mathbf{s}_t$
    Update $G' = (N \setminus N_h, E')$ {Exclude nodes $N_h$}
    Compute shortest path using A* on $G'$
    **if** human activity detected **then**
      **if** dynamic adjustment possible **then**
        $\mathbf{s}_2' = \arg\min_{\mathbf{s}}\{c(\mathbf{s}_1, \mathbf{s}) \mid \mathbf{s} \neq h_2\}$ {Dynamic interaction strategy: find new state avoiding human activity}
      **else**
        Reroute to $\mathbf{s}_t'$ if $d(\mathbf{s}_t, h_t) < \delta$ {Conservative avoidance strategy: reroute if within avoidance threshold}
      **end if**
    **end if**
    $\mathcal{C}(\mathbf{s}_t, h_t) = 1$ if $d(\mathbf{s}_t, h_t) < \epsilon$ {Collision avoidance strategy: mark collision if too close}
    Execute $a_t$
  **end while**

---

## C.3  Algorithm to Construct VLN-DT

The pseudocode for the structure and training of VLN-DT, presented in a `Python`-style format, is summarized in algorithm 2. Note that we use the pseudocode template from [8]. The VLN-DT model takes as input the returns-to-go ($R$), instructions ($I$), current observations ($\Theta$), actions ($a$), and timesteps ($t$). The key components of the model include the transformer with causal masking, embedding layers for state, action, and returns-to-go, a learned episode positional embedding, a cross-modality fusion module, BERT layers for language embedding, a ResNet-152 feature extractor for visual embedding, and a linear action prediction layer.

The main `VLNDecisionTransformer` function computes the BERT embedding for instructions and the CNN feature map for visual observations. These embeddings are then fused using the cross-modality fusion module to obtain a unified representation. Positional embeddings are computed for each timestep and added to the token embeddings for state, action, and returns-to-go. The resulting interleaved tokens are passed through the transformer to obtain hidden states, from which the hidden states corresponding to action prediction tokens are selected. Finally, the next action is predicted using the linear action prediction layer.

During the training loop, the VLN-DT model is trained using a cross-entropy loss for discrete actions. The optimizer is used to update the model parameters based on the computed gradients. In the evaluation loop, the target return is set (e.g., expert-level return), and the model generates actions autoregressively. At each timestep, the next action is sampled using the VLN-DT model, and the environment is stepped forward to obtain a new observation and reward. The returns-to-go are updated, and new tokens are appended to the sequence while maintaining a context length of $K$.

## C.4  Different Reward Types for VLN-DT

To train the VLN-DT agent effectively, we define three distinct reward types that capture different aspects of the navigation task and encourage desirable behaviors.

**Target Reward.** The target reward is defined as follows:

$$r_t^{\text{target}} = \begin{cases} 5, & \text{if } d(s_t, \text{target}) \leq threshold \\ -5, & \text{otherwise} \end{cases}$$

This reward type incentivizes the agent to reach the target location within a specified distance threshold. If the agent stops within a distance $threshold$ from the target, it receives a positive reward of 5. Otherwise, if the agent fails to reach the target or stops far from it, a negative reward of -5 is given. This reward encourages the agent to navigate accurately and reach the desired destination.

**Distance Reward.** The distance reward is defined as follows:

$$r_t^{\text{distance}} = \begin{cases} 1, & \text{if } d(s_t, \text{target}) < d(s_{t-1}, \text{target}) \\ -0.1, & \text{otherwise} \end{cases}$$

The distance reward aims to encourage the agent to move closer to the target location with each step. If the agent's current state $s_t$ is closer to the target than its previous state $s_{t-1}$, it receives a positive reward of 1. On the other hand, if the agent moves away from the target, a small penalty of -0.1 is applied. This reward type helps guide the agent towards the target and promotes efficient navigation.

**Human Reward.** The human reward is defined as follows:

$$r_t^{\text{human}} = \begin{cases} 0, & \text{if no collision with human} \\ -2, & \text{if collision occurs} \end{cases}$$

The human reward is designed to penalize the agent for colliding with humans. If the agent navigates without colliding with any humans, it receives a neutral reward of 0. However, if a collision with a human occurs, the agent incurs a significant penalty of -2. This reward type encourages the agent to navigate safely and avoid collisions, promoting socially-aware navigation behaviors.

By incorporating these three reward types, the VLN-DT agent is trained to balance multiple objectives: reaching the target location accurately, moving closer to the target with each step, and avoiding collisions with humans. The target reward provides a strong incentive for the agent to reach the desired destination, while the distance reward encourages efficient navigation by rewarding the agent

for making progress towards the target. The human reward ensures that the agent learns to navigate in a socially-aware manner, prioritizing the safety of humans in the environment. During training, these rewards are combined to form the overall reward signal that guides the learning process of the VLN-DT agent. By optimizing its behavior based on these rewards, the agent learns to navigate in the presence of human activities, aligning with the goals of the HA-VLN task.

---

**Algorithm 2** VLN-DT Structure and Training Pseudocode (for discrete actions)

```
# R, a, t: returns-to-go, actions, or timesteps
# I, Theta, instructions, current observations
# transformer: transformer with causal masking (GPT)
# embed_s, embed_a, embed_R: linear embedding layers
# embed_t: learned episode positional embedding
# modality_fuse: cross modality fusion module
# bert_embed: bert layers
# cnn: Resnet152 feature extractor
# pred_a: linear action prediction layer

# main model
def VLNDecisitionTransformer(R, I, Theta, a, t):
    # compute bert embedding and image feature map
    e = bert_embed(I)
    c = cnn(Theta)

    # compute unified representation
    s = modality_fuse(e, c)

    # compute embeddings for tokens in transformer
    pos_embedding = embed_t(t)  # per-timestep , not per-token
    s_embedding = embed_s(s) + pos_embedding
    a_embedding = embed_a(a) + pos_embedding
    R_embedding = embed_R(R) + pos_embedding

    # interleave tokens as (R_1, s_1, a_1, ..., R_K, s_K)
    input_embeds = stack(R_embedding, s_embedding, a_embedding)

    # use transformer to get hidden states
    hidden_states = transformer(input_embeds=input_embeds)

    # select hidden states for action prediction tokens
    a_hidden = unstack(hidden_states).actions

    # predict action
    return pred_a(a_hidden)

# training loop
for (R, I, Theta, t) in dataloader:  # dims: (batch_size, K, dim)
    a_preds = VLNDecisionTransformer(I, Theta, a, t)
    loss = ce(a_preds, a)  # cross entropy loss for continuous actions
    optimizer.zero_grad(); loss.backward(); optimizer.step()

# evaluation loop
target_return = 1  # for instance, expert-level return
R, I, Theta, a, t, done = [target_return], [env.reset()], [], [1],
    False
while not done:  # autoregressive generation/sampling
    # sample next action
    action = VLNDecisionTransformer(R, I, Theta, a, t)[-1]
    I, new_Theta, r, done, _ = env.step(action)

    # append new tokens to sequence
    R = R + [R[-1] - r]  # decrement returns-to-go with reward
    Theta, a, t = Theta + [new_Theta], a + [action], t + [len(R)]
    R, Theta, a, t = R[-K:], ...  # only keep context length of K
```

---

# D Experiment Details

## D.1 Evaluation Protocol

In HA-VLN, we construct a fair and comprehensive assessment of the agent's performance by incorporating critical nodes in the evaluation metrics. To help better understand the new evaluation metrics defined in the main text, the *original metrics* before such an update are as follows:

**Total Collision Rate (TCR).** The Total Collision Rate measures the overall frequency of the agent colliding with any obstacles or areas within a specified radius. It is calculated as the average number of collisions per navigation instruction, taking into account the presence of critical nodes. The formula for TCR is given by:

$$\text{TCR} = \frac{\sum_{i=1}^{L} c_i}{L}$$

where $c_i$ represents the number of collisions within a 1-meter radius in navigation instance $i$, and $L$ denotes the total number of navigation instances. By considering collisions in the vicinity of critical nodes, TCR provides a comprehensive assessment of the agent's ability to navigate safely in the presence of obstacles and important areas.

**Collision Rate (CR).** The Collision Rate assesses the proportion of navigation instances that experience at least one collision, taking into account the impact of critical nodes. It is calculated using the following formula:

$$\text{CR} = \frac{\sum_{i=1}^{L} \min(c_i, 1)}{L}$$

where $\min(c_i, 1)$ ensures that any instance with one or more collisions is counted only once. By focusing on the occurrence of collisions rather than their frequency, CR provides a complementary perspective on the agent's navigation performance, highlighting the proportion of instructions that encounter collisions in the presence of critical nodes.

**Navigation Error (NE).** The Navigation Error measures the average distance between the agent's final position and the target location across all navigation instances, considering the influence of critical nodes. It is calculated using the following formula:

$$\text{NE} = \frac{\sum_{i=1}^{L} d_i}{L}$$

where $d_i$ represents the distance error in navigation instance $i$. By taking into account the proximity to critical nodes when calculating the distance error, NE provides a more nuanced evaluation of the agent's navigation accuracy, penalizing deviations that occur near important areas.

**Success Rate (SR).** The Success Rate calculates the proportion of navigation instructions completed successfully without any collisions, considering the presence of critical nodes. It is determined using the following formula:

$$\text{SR} = \frac{\sum_{i=1}^{L} \mathbb{I}(c_i = 0)}{L}$$

where $\mathbb{I}(c_i = 0)$ is an indicator function equal to 1 if there are no collisions in the navigation instance $i$ and 0 otherwise. By requiring the absence of collisions for a successful navigation, SR provides a stringent evaluation of the agent's ability to complete instructions safely.

The Total Collision Rate (TCR) and Collision Rate (CR) capture different aspects of collision avoidance, with TCR measuring the overall frequency of collisions and CR focusing on the proportion of instructions affected by collisions. The Navigation Error (NE) evaluates the agent's accuracy in reaching the target location, while the Success Rate (SR) assesses the agent's ability to complete instructions without any collisions.

By leveraging these metrics, researchers can gain a holistic understanding of the agent's performance in the HA-VLN task, identifying strengths and weaknesses in navigation safety, accuracy, and success. Compared to the original metrics, our updated comprehensive evaluation framework enables the development and comparison of agents that can effectively navigate in the presence of critical nodes, paving the way for more robust and reliable human-aware navigation systems. This approach also ensures that the evaluation of agents is rigorous and reflects real-world scenarios where navigating in human-populated environments presents significant challenges.

Table 10: Impact of Critical Nodes on Agent Navigation Performance on HA-VLN. The table compares the performance of the Airbert agent excluding the impact of critical nodes (*w/o critical nodes*) and including the impact of critical nodes (*w/ critical nodes*). The results show that ignoring critical nodes can overestimate the *human perception* ability of agents.

| Env Name | w/ critical nodes | | w/o critical nodes | | Difference | |
|---|---|---|---|---|---|---|
| | TCR | CR | TCR | CR | TCR | CR |
| **Validation Seen** | 0.191 | 0.644 | 0.146 | 0.515 | **+30.8%** | **+25.0%** |
| **Validation Unseen** | 0.281 | 0.764 | 0.257 | 0.689 | **+9.3%** | **+10.9%** |

## D.2 Evaluating the Impact of Critical Nodes

To assess the impact of critical nodes on agent performance in the HA-VLN task, we trained the Airbert agent using a panoramic action space and sub-optimal expert supervision. Tab. 10 presents the *human-aware* performance difference between including the impact of critical nodes (*w/ critical nodes*) and excluding their impact (*w/o critical nodes*).

The results reveal that including the impact of critical nodes in the HA-VLN task leads to an underestimation of the agent's ability to navigate in realistic environments (Sim2Real ability). Specifically, when critical nodes are excluded from the evaluation, both the Total Collision Rate (TCR) and Collision Rate (CR) show considerable improvements of 30.8% and 25.0%, respectively, in the validation seen environment. This suggests that ignoring the impact of critical nodes can lead to an overestimation of the agent's human perception and navigation capabilities.

The observed differences in performance highlight the importance of considering critical nodes when assessing an agent's navigational efficacy in the HA-VLN task. Critical nodes represent crucial points in the navigation environment where the agent's behavior and decision-making are particularly important, such as narrow passages, doorways, or areas with high human activity. By including the impact of critical nodes, we obtain a more realistic and accurate evaluation of the agent's ability to navigate safely and efficiently in the presence of human activities.

Furthermore, the results underscore the significance of critical nodes in bridging the gap between simulated and real-world environments (Sim2Real gap). By incorporating the impact of critical nodes during training and evaluation, we can develop agents that are better equipped to handle the challenges and complexities encountered in real-world navigation scenarios.

In light of these findings, we argue that excluding the impact of critical nodes leads to a fairer and more comprehensive assessment of an agent's navigational performance on the HA-VLN task. By focusing on the agent's behavior and decision-making at critical nodes, we can obtain insights into its ability to perceive and respond to human activities effectively.

Therefore, in the experiments presented in this work, we exclude the impact of critical navigation nodes to ensure a rigorous and unbiased evaluation of the agents' performance on the HA-VLN task. This approach allows us to accurately assess the agents' capabilities in navigating dynamic, human-aware environments and provides a solid foundation for developing robust and reliable navigation systems that can operate effectively in real-world settings.

## D.3 Evaluating the Oracle Performance

To evaluate the performance of the oracle agents in the HA-VLN task, we conducted a comparative analysis between the sub-optimal expert and the optimal expert. Tab. 11 presents the results of this evaluation, providing insights into the strengths and limitations of each expert agent.

The optimal expert achieves the highest Success Rate (SR) of 100% in both seen and unseen environments, demonstrating its ability to navigate effectively and reach the target destination. However, this high performance comes at the cost of increased Total Collision Rate (TCR) and Collision Rate (CR). In the validation unseen environment, the optimal expert exhibits a staggering 800% increase in TCR and a 1700% increase in CR compared to the sub-optimal expert. These substantial increases in collision-related metrics indicate that the optimal expert prioritizes reaching the goal over avoiding collisions with humans and obstacles.

Table 11: Impact of Expert Quality on Ground Truth (oracle) in HA-VLN. The table compares the performance of expert agents. The results indicate that the sub-optimal expert provides weak supervision signals for navigation by balancing NE, TCR, CR, and SR.

| Expert Agent | Validation Seen | | | | Validation Unseen | | | |
|---|---|---|---|---|---|---|---|---|
| | NE ↓ | TCR ↓ | CR ↓ | SR ↑ | NE ↓ | TCR ↓ | CR ↓ | SR ↑ |
| Sub-optimal$_{oracle}$ | 0.67 | 0.04 | 0.04 | 0.89 | 0.62 | 0.01 | 0.01 | 0.91 |
| Optimal$_{oracle}$ | 0.00 | 0.14 | 0.22 | 1.00 | 0.00 | 0.09 | 0.18 | 1.00 |
| Difference | -0.67 | +0.10 | +0.18 | +0.11 | -0.62 | +0.08 | +0.17 | +0.09 |
| Percentage Change | -100.0% | +250.0% | +450.0% | +12.4% | -100.0% | +800.0% | +1700.0% | +9.9% |

On the other hand, the sub-optimal expert provides a more balanced approach to navigation. Although its SR is slightly lower than the optimal expert by 11.0% in seen environments and 9.9% in unseen environments, the sub-optimal expert achieves significantly lower TCR and CR. This suggests that the sub-optimal expert strikes a better balance between navigation efficiency and human-aware metrics, making it more suitable for real-world applications.

The sub-optimal expert's performance can be attributed to its ability to navigate while considering the presence of humans and obstacles in the environment. By prioritizing collision avoidance and maintaining a safe distance from humans, the sub-optimal expert provides a more practical approach to navigation in dynamic, human-populated environments. This is particularly important in real-world scenarios where the safety and comfort of humans are paramount.

Moreover, the sub-optimal expert's balanced performance across navigation-related and human-aware metrics makes it an ideal candidate for providing weak supervision signals during the training of navigation agents. By learning from the sub-optimal expert's demonstrations, navigation agents can acquire the necessary skills to navigate efficiently while being mindful of human presence and potential collisions.

The oracle performance analysis highlights the importance of considering both navigation efficiency and human-aware metrics when evaluating expert agents and training navigation agents. While the optimal expert excels in reaching the target destination, its high collision rates limit its practicality in real-world scenarios. The sub-optimal expert, on the other hand, provides a more balanced approach, achieving reasonable success rates while minimizing collisions with humans and obstacles. By incorporating the sub-optimal expert's demonstrations during training, navigation agents can learn to navigate effectively and safely in complex, human-populated environments, bridging the gap between simulation and real-world applications (i.e., Sim2Real Challenges).

### D.4 Evaluating on Real-World Robots

**Robot Setup**. To validate the performance of our navigation agents in real-world scenarios, we conducted experiments using a Unitree GO1-EDU quadruped robot. Fig. 15 provides a detailed visual representation of the robot and its key components. The robot is equipped with a stereo fisheye camera mounted on its head, which captures RGB images with a 180-degree field of view. To align with the agent's Ergonomic Action Space setup, we cropped the central 60 degrees of the camera's field of view and used it as the agent's visual input. It is important to note that our approach only utilizes monocular images from the fisheye camera.

In addition to the camera, the robot is equipped with an ultrasonic distance sensor located beneath the fisheye camera. This sensor measures the distance between the robot and humans, enabling the calculation of potential collisions. An Inertial Measurement Unit (IMU) is also integrated into the robot to capture its position and orientation during navigation.

To deploy our navigation agents, the robot is equipped with an NVIDIA Jetson TX2 AI computing device. This high-performance computing module handles the computational tasks required by the agent, such as receiving images and inferring the next action command. The agent's action commands are then executed by the Motion Control Unit, which is implemented using a Raspberry Pi 4B. This unit sets the robot in a high-level motion mode, allowing it to directly execute movement commands such as "turn left" or "move forward." The minimum movement distance is set to 0.5m, and the turn angle is set to 45 degrees. Throughout the robot's movements, the IMU continuously tracks the motion to ensure that the rotations and forward movements align with the issued commands.

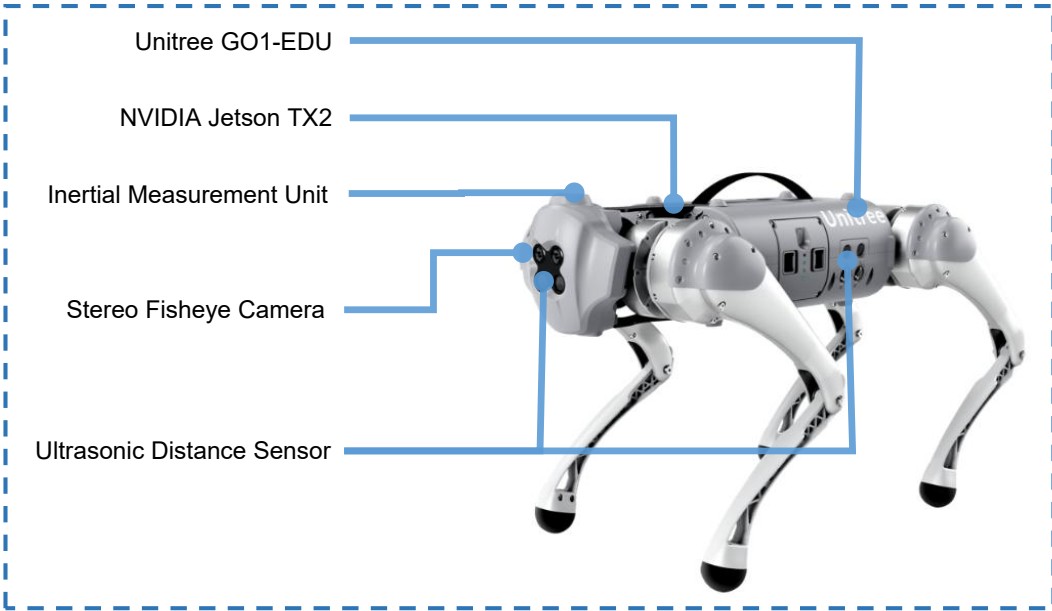

Figure 15: Real-world robot used in our experiments. The robot is Unitree GO1-EDU, a quadruped robot equipped with an NVIDIA Jetson TX2 high-performance computing module for handling computational tasks. The robot features an Inertial Measurement Unit (IMU) for measuring acceleration and rotational speed, a Stereo Fisheye Camera for wide-angle perception of its surroundings, and an Ultrasonic Distance Sensor for measuring the distance between the robot and obstacles.

**Visual Results of Demonstration**. To showcase the real-world performance of our navigation agents, we provide visual results of the robot navigating in various office environments. Fig. 16 demonstrates the robot successfully navigating an office environment without human presence. The figure presents the instruction given to the robot, the robot's view captured by the fisheye camera, and a third-person view of the robot's navigation.

In Fig. 17, we present an example of the robot navigating in an office environment with human activity. The robot observes humans in its surroundings, adjusts its path accordingly, circumvents the humans, and ultimately reaches its designated destination. This showcases the robot's ability to perceive and respond to human presence while navigating.

However, it is important to acknowledge that the robot's performance is not infallible. Fig. 18 illustrates a scenario where the robot collides with a human, even in the same environment. This collision occurs when the human's status changes unexpectedly, leading to a mission failure. This example highlights the challenges and limitations of real-world navigation in dynamic human environments. To provide a more view of the robot's navigation capabilities, we have made the complete robot navigation video available on our project website. This video showcases various scenarios and provides a deeper understanding of the robot's performance in real-world settings.

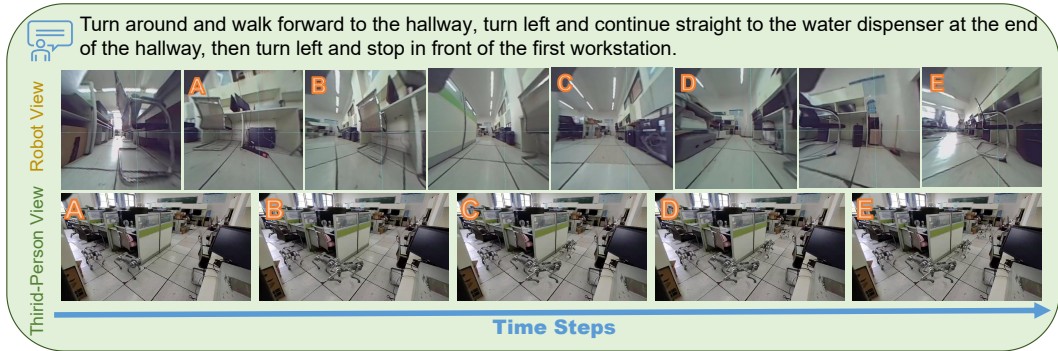

Figure 16: Example of a robot successfully navigating in a real environment without human presence. The figure presents the instruction given to the robot (top), the robot's view captured by the fisheye camera (middle), and a third-person view of the robot's navigation (bottom). [Zoom in to view]

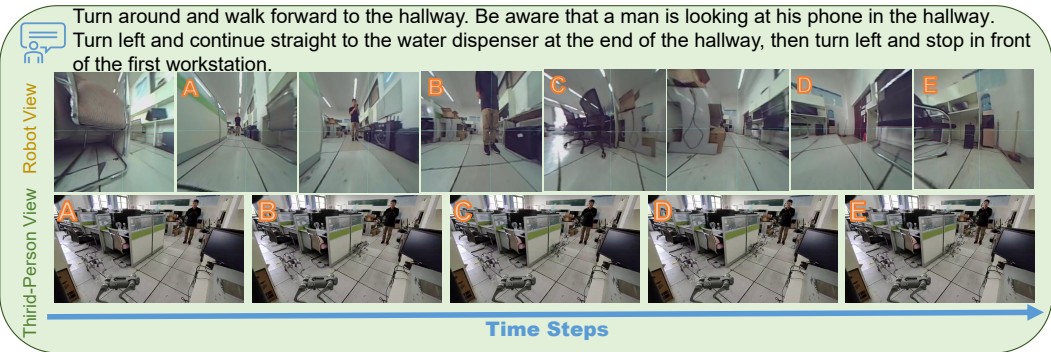

Figure 17: Example of a robot successfully navigating in a real environment with human activity. The robot observes humans, adjusts its path, circumvents them, and reaches its designated destination. [Zoom in to view]

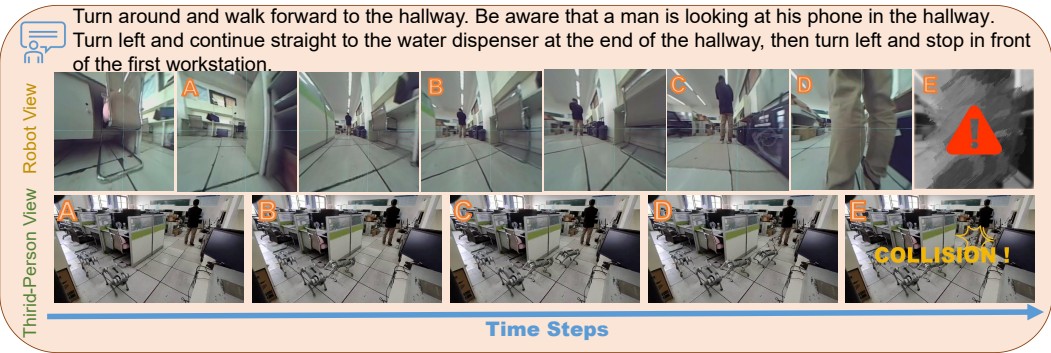

Figure 18: Example of robot navigation failures in real environments with human presence. The robot collides with a human when their status changes unexpectedly, leading to a mission failure. [Zoom in to view]

