# OpenReview forum: "Human-Aware Vision-and-Language Navigation: Bridging Simulation to Reality with Dynamic Human Interactions"
_NeurIPS.cc/2024/Datasets_and_Benchmarks_Track — NeurIPS 2024 Track Datasets and Benchmarks Spotlight_

### Official Review · Reviewer_Np5K · 2024-07-22

**Rating:** 8
**Confidence:** 4
**Correctness:** the claims made in the submission are…
**Clarity:** the paper is well written

**Review:**

The paperoffers a substantial contribution to the field of Vision-and-Language Navigation (VLN) by introducing dynamic human activities into the navigation framework, which significantly enhances the real-world applicability of VLN systems. The research is of high quality and demonstrates originality, particularly through the development of the Human-Aware 3D (HA3D) simulator and the Human-Aware Room-to-Room (HA-R2R) dataset. These tools effectively bridge the gap between simulated environments and real-world scenarios, enabling agents to navigate dynamically populated spaces. The methodology is clearly and comprehensively detailed, particularly in the innovative use of egocentric action spaces and the novel training strategies for the proposed navigation agents, VLN-CM and VLN-DT. The corresponding pros and cons can be seen in Stregths and Oppotunities for Improvement Section.

**Strengths:**

1. Extend traditional VLN by incorporating dynamic human activities, addressing the gap between simulation and real-world scenarios, which is crucial for practical applications.
2.  Introduce new evaluation metrics that consider human activities, providing a more accurate assessment of an agent's performance in real-world conditions.
3. The paper provides valuable insights and benchmarks for future research in embodied AI and Sim2Real transfer

**Additional Feedback:**

No

**Documentation:**

The paper provides substantial details on the data collection and organization processes, and a link on project website.

**Limitations:**

See above

**Opportunities For Improvement:**

1. Extending the HA3D and HA-R2R datasets to include outdoor scenarios could provide a more comprehensive training environment, enabling agents to navigate a wider variety of real-world settings and further improving their versatility and applicability.
2. Some figures in the paper, such as Fig 3(c), Fig 4, and Fig 5, have fonts that are too small, making them difficult to read. Improving the clarity and readability of these figures would enhance the overall presentation and comprehension of the paper.

**Relation To Prior Work:**

The paper clearly differentiates by emphasizing three key advancements. Unlike prior methods that predominantly used panoramic action spaces, the HA-VLN framework employs an egocentric action space. It also addresses the lack of dynamic human interactions in earlier simulators by introducing the HA3D simulato. Lastly, the HA-VLN framework uses sub-optimal expert supervision instead of the optimal expert guidance relied upon by traditional VLN models.

**Summary And Contributions:**

This paper introduces Human-Aware VLN  to improve the real-world applicability of VLN systems by incorporating dynamic human activities. The authors present the HA3D simulator and the HA-R2R dataset, which enhance traditional VLN tasks by integrating 3D human motion models and activity descriptions.

---

> ### Author Rebuttal · Authors · 2024-08-17
>
> Thank you for your positive evaluation of our work. We are pleased that the significance and contributions of our paper were clearly conveyed and appreciated.
>
>
> ---
>
> **Q1: The focus on indoor environments is well-justified, but have you considered expanding the HA3D and HA-R2R datasets to include outdoor scenarios to further improve the versatility and applicability of your approach?**
>
> **A1:**  Thank you for this insightful suggestion. Our current work focuses on indoor environments, which aligns with the established Room-to-Room (R2R) framework commonly used in Vision-and-Language Navigation (VLN) research [1][2][3][4]. This focus allowed us to make meaningful comparisons with prior work and address the unique challenges of dynamic human interactions in confined spaces.
>
> We agree that expanding to outdoor scenarios would enhance the versatility and applicability of our approach. In future work, we plan to extend our datasets and simulations to include outdoor environments, such as urban streets, parks, and mixed indoor-outdoor spaces. This expansion will involve capturing dynamic human activities under various environmental conditions, such as different lighting and weather, which are critical for robust Sim2Real transfer. We believe these additions will contribute significantly to the field by providing a more comprehensive benchmark for evaluating VLN systems in diverse real-world settings.
>
> ---
>
> **Q2: Some of the figures in the paper, particularly Fig. 3(c), Fig. 4, and Fig. 5, have fonts that are too small, which might make them difficult to read. Could improving the clarity of these figures enhance the overall presentation and comprehension of the paper?**
>
> **A2:**  We appreciate your attention to detail regarding the visual presentation of our work. In response to your feedback, we have revised the figures in question. We increased the font sizes and improved the overall clarity of Fig. 3(c), Fig. 4, and Fig. 5, which are now updated as Fig. 1, Fig. 2, and Fig. 3 in the revised manuscript. We believe these changes will significantly enhance the readability and overall comprehension of our visual data.
>
> ---
>
> **References:**
>
> [1] (https://arxiv.org/abs/1711.07280) Anderson, Peter, et al. "Vision-and-language navigation: Interpreting visually-grounded navigation instructions in real environments." Proceedings of the IEEE conference on computer vision and pattern recognition. 2018.
>
> [2] (https://www.ecva.net/papers/eccv_2020/papers_ECCV/papers/123730103.pdf) Krantz, Jacob, et al. "Beyond the nav-graph: Vision-and-language navigation in continuous environments." Computer Vision–ECCV 2020: 16th European Conference, Glasgow, UK, August 23–28, 2020, Proceedings, Part XXVIII 16. Springer International Publishing, 2020.
>
> [3] (https://proceedings.mlr.press/v155/anderson21a/anderson21a.pdf) Anderson, Peter, et al. "Sim-to-real transfer for vision-and-language navigation." Conference on Robot Learning. PMLR, 2021.
>
> [4] (https://arxiv.org/pdf/2407.07035) Zhang, Yue, et al. "Vision-and-Language Navigation Today and Tomorrow: A Survey in the Era of Foundation Models." arXiv preprint arXiv:2407.07035 (2024).
>
> ---
>
> We hope these clarifications address your concerns. If there are any additional questions or further clarifications needed, please do not hesitate to reach out. We are committed to continuously improving our work and value your insightful feedback.

---

### Comment · Area_Chair_qE6s · 2024-09-04
**A review report.**

Summary and contributes:

The paper introduces a new framework for Vision-and-Language Navigation by incorporating dynamic human activities into the environment, enhancing its real-world applicability. The authors propose the HA3D simulator, which integrates human activities with the Matterport3D dataset, and the HA-R2R dataset, extending existing datasets with human activity descriptions. They develop two navigation agents VLN-CM and VLN-DT and evaluate them using new metrics that consider human interactions. The study highlights the need for more robust navigation agents for human-populated environments and demonstrates the potential of these methods in real-world applications.



Reviews:

See strengths and weaknesses (in "Opportunities for Improvement") below.



Strengths:

1.This paper introduces the HA-VLN task, which extends traditional VLN by incorporating dynamic human activities and relaxing certain assumptions.

2.The HA3D simulator and HA-R2R dataset introduce a novel, dynamic environment with human activities to support HA-VLN research and enable the development of robust navigation agents.

3.The paper provides benchmarks and insights for future research on embodied AI and Sim2Real transfer.



Rating：7



Opportinuities for improvement:

1.The study is limited to indoor environments and does not consider more complex outdoor or mixed scenarios; it also does not cover dynamic situations involving multiple individuals simultaneously.

2.Figures like 3(c) and 4 have fonts that are too small, making them hard to read.



Confidence:3



Limitations:

See above



Correctness:

the claims made in the paper are generally correct



Clarity:

Yes



Relation to prior work:

This paper advances VLN research by incorporating dynamic human activities into navigation tasks, addressing limitations of static environments and optimal supervision in previous works. The HA-VLN task, supported by the HA3D simulator and HA-R2R dataset, offers a more realistic setting for developing robust, human-aware navigation agents.



Documentation:

The paper provides substantial details on the data collection processes and includes a link for access.



Ethics: No

Flag For Ethics Review: 2



Additional Feedback: N/A

---

> ### Author Response · Authors · 2024-09-05
> **Rebuttal by Authors-1**
>
> Thank you for your positive evaluation of our work. We are pleased that the significance and contributions of our paper were clearly conveyed and appreciated.
>
> **Q1: The study is limited to indoor environments and does not consider more complex outdoor or mixed scenarios.**
>
> **A1:** We appreciate the reviewer's observation regarding the current scope of our study being limited to indoor environments. We acknowledge the importance of extending our research to outdoor and mixed scenarios. Our decision to focus on indoor environments in the current version was based on the following considerations:
>
> - **Alignment with Established VLN Tasks:** Our primary goal was to align our work with other prominent Visual Language Navigation (VLN) tasks. Notable studies in this field, such as [1], [2], [3], and [4], have consistently used indoor environments as their primary setting for VLN tasks.
>
> - **Challenges with Outdoor VLN:** Most advanced outdoor visual language navigation technologies are derived from autonomous driving research [6]. These typically require long-distance path planning [5], which presents challenges for achieving fine-grained alignment between short instructions and navigation processes. This approach would deviate from the original intent of VLN tasks [1], which focus on more granular navigation in confined spaces.
>
> We recognize the value in expanding our research to include outdoor and mixed environments. In our future work, we plan to extend our dataset and simulations to encompass:
>
> 1. **Dynamic Scene Elements:** We will capture various environmental conditions, including different lighting and weather scenarios, as well as incorporate dynamic human activities. This is crucial for the robustness of Sim2Real transfer.
> 2. **Diverse Scenarios:** By introducing a wider range of environments, including city streets, parks, and mixed indoor-outdoor spaces, we aim to enhance the versatility and applicability of our HA-VLN system.
>
> These extensions will significantly broaden the scope of our research, addressing the current limitation while maintaining the core principles of VLN tasks. We believe this expansion will contribute to a more comprehensive understanding of visual language navigation across diverse real-world scenarios.
>
> **Q2: It also does not cover dynamic situations involving multiple individuals simultaneously.**
>
> **A2:** We appreciate the reviewer's concern regarding dynamic situations involving multiple individuals simultaneously. We would like to clarify that our HA-VLN system does, in fact, address such scenarios. We have considered and addressed the following two main types of multi-person activity scenarios:
>
> 1. **Multiple individuals active in the agent's field of view, but not gathered together.**
> 2. **Multiple individuals active in the agent's field of view and gathered together.**
>
> #### Scenario 1: Dispersed Individuals
>
> Our agents frequently encounter multiple people during HA-VLN navigation. As evidenced in Figures 11 and 13 of our paper, there are instances where multiple individuals are simultaneously present in the agent's view. Specifically:
>
> - In Figure 11, the two images in the lower right corner demonstrate this scenario.
> - In Figure 13, the last image also clearly shows multiple people in the agent's field of view.
>
> These examples illustrate that our system is capable of handling situations with multiple, dispersed individuals.
>
> #### Scenario 2: Gathered Individuals
>
> We considered this scenario during the initial design of our simulator. However, we determined that it did not significantly impact our mission design for the following reasons:
>
> 1. In our VLN mission, the average distance between each waypoint is 2.5 meters.
> 2. Most indoor multi-person human activities typically occur within this range.
> 3. Consequently, the impact on navigation routes is essentially the same whether encountering a single person or a small group of people within this limited space.
>
> Given these factors, our current approach effectively addresses both single and multiple human activities in indoor environments without requiring separate handling for gathered individuals. While we appreciate the reviewer's attention to this detail, we believe our HA-VLN system adequately addresses dynamic situations involving multiple individuals, both dispersed and gathered, within the context of our navigation tasks.

---

> > ### Author Response · Authors · 2024-09-05
> > **Rebuttal by Authors-2**
> >
> > **Q3: Figures like 3(c) and 4 have fonts that are too small, making them hard to read.**
> >
> > **A3:** We have addressed this concern in our revised manuscript. We have significantly increased the font sizes in Figures 3(c), 4, and 5. We've made comprehensive improvements to enhance the overall clarity and visual presentation of these figures.
> >
> > ### **References:**
> >
> > [1] Anderson, Peter, et al. "Vision-and-language navigation: Interpreting visually-grounded navigation instructions in real environments." Proceedings of the IEEE conference on computer vision and pattern recognition. 2018. [Link](https://arxiv.org/abs/1711.07280)
> >
> > [2] Krantz, Jacob, et al. "Beyond the nav-graph: Vision-and-language navigation in continuous environments." Computer Vision–ECCV 2020: 16th European Conference, Glasgow, UK, August 23–28, 2020, Proceedings, Part XXVIII 16. Springer International Publishing, 2020. [Link](https://www.ecva.net/papers/eccv_2020/papers_ECCV/papers/123730103.pdf)
> >
> > [3] Anderson, Peter, et al. "Sim-to-real transfer for vision-and-language navigation." Conference on Robot Learning. PMLR, 2021. [Link](https://proceedings.mlr.press/v155/anderson21a/anderson21a.pdf)
> >
> > [4] Zhang, Yue, et al. "Vision-and-Language Navigation Today and Tomorrow: A Survey in the Era of Foundation Models." arXiv preprint arXiv:2407.07035 (2024). [Link](https://arxiv.org/pdf/2407.07035)
> >
> > [5] Zhang, Yue, Ziqiao Ma, Jialu Li, Yanyuan Qiao, Zun Wang, Joyce Chai, Qi Wu, Mohit Bansal, and Parisa Kordjamshidi. "Vision-and-Language Navigation Today and Tomorrow: A Survey in the Era of Foundation Models." arXiv.org, July 9, 2024. [Link](https://arxiv.org/abs/2407.07035v1)
> >
> > [6] Shah, Dhruv, Ajay Sridhar, Nitish Dashora, Kyle Stachowicz, Kevin Black, Noriaki Hirose, and Sergey Levine. "ViNT: A Foundation Model for Visual Navigation." arXiv, October 24, 2023. [Link](https://doi.org/10.48550/arXiv.2306.14846)
> >
> > ---
> >
> > We hope these clarifications address your concerns. If there are any additional questions or further clarifications needed, please do not hesitate to reach out.

---

### Decision · Program_Chairs · 2024-09-26

**Decision:**

Accept (Spotlight)

**Comment:**

The paper introduces a new framework for Vision-and-Language Navigation by incorporating dynamic human activities into the environment. After repeated communication between the author and the reviewers, the relevant doubts and uncertainties were resolved. Ultimately, based on various factors, including innovation and rigor, I have determined that this paper can be accepted.